# Implicit Regularization of Sharpness-Aware Minimization for Scale-Invariant Problems

**Bingcong Li**        **Liang Zhang**        **Niao He**

Department of Computer Science
ETH Zurich, Switzerland
`{bingcong.li, liang.zhang, niao.he}@inf.ethz.ch`

## Abstract

Sharpness-aware minimization (SAM) improves generalization of various deep learning tasks. Motivated by popular architectures such as LoRA, we explore the implicit regularization of SAM for scale-invariant problems involving two groups of variables. Instead of focusing on commonly used sharpness, this work introduces a concept termed *balancedness*, defined as the difference between the squared norm of two variables. This allows us to depict richer global behaviors of SAM. In particular, our theoretical and empirical findings reveal that i) SAM promotes balancedness; and ii) the regularization on balancedness is *data-responsive* – outliers have stronger impact. The latter coincides with empirical observations that SAM outperforms SGD in the presence of outliers. Leveraging the implicit regularization, we develop a resource-efficient SAM variant, balancedness-aware regularization (BAR), tailored for scale-invariant problems such as finetuning language models with LoRA. BAR saves $95\%$ computational overhead of SAM, with enhanced test performance across various tasks on RoBERTa, GPT2, and OPT-1.3B.

## 1 Introduction

Sharpness-aware minimization (SAM) is emerging as an appealing optimizer, because it enhances generalization performance on various downstream tasks across vision and language applications (Foret et al., 2021; Chen et al., 2022; Bahri et al., 2022). The success of SAM is typically explained using its implicit regularization (IR) toward a flat solution (Wen et al., 2023a).

However, existing results only characterize sharpness/flatness near *local* minima (Wen et al., 2023a). Little is known about early convergence, despite its crucial role in SAM's implicit regularization (Agarwala and Dauphin, 2023). In addition, theoretical understanding of SAM highly hinges upon the existence of positive eigenvalues of Hessians (Wen et al., 2023a), leaving gaps in nonconvex scenarios where the Hessian can be negative definite. The limitations above lead to our first question (**Q1**): *can we broaden the scope of implicit regularization to depict global behaviors in SAM?*

Moreover, scenarios where SAM popularizes often involve certain form of data anomalies, such as outliers and large data variance. SAM has provable generalization benefits on sparse coding problems in the small signal-to-noise ratio (SNR) regime (Chen et al., 2023). Remarkable performance of SAM is also observed under distributional shifts, e.g., domain adaptation (Wang et al., 2023), meta-learning (Abbas et al., 2022), and transfer learning in language models (Bahri et al., 2022; Sherborne et al., 2023). Evidences above motivate our second question (**Q2**): *can implicit regularization of SAM reflect its enhanced performance under data anomalies?*

This work answers both Q1 and Q2 within a class of *scale-invariant* problems. The focus on scale-invariance is motivated by its prominence in deep learning architectures. Consider variables $\mathbf{x} \in \mathbb{R}^{d_1}$

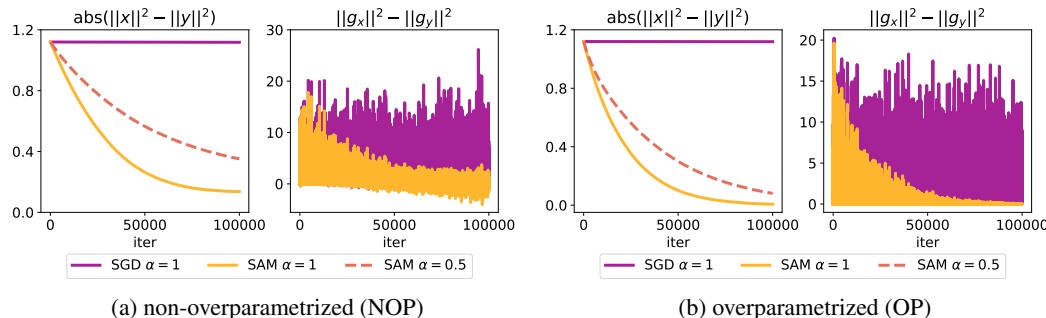

Figure 1: Implicit regularization of SAM on balancedness. The losses for NOP and OP are $\mathbb{E}[\|\mathbf{x}\mathbf{y}^\top - (\mathbf{A} + \alpha\mathbf{N})\|^2]$ and $\mathbb{E}[\|\mathbf{x}^\top\mathbf{y} - (a + \alpha n)\|^2]$, respectively. Here, $\mathbf{A}$ is the ground truth matrix, $\mathbf{N}$ is the Gaussian noise, and $\alpha$ controls the SNR. Left of (a) and (b): $|\|\mathbf{x}_t\|^2 - \|\mathbf{y}_t\|^2|$ vs. iteration. Right of (a) and (b): $|\|\mathbf{g}_{\mathbf{x}_t}\|^2 - \|\mathbf{g}_{\mathbf{y}_t}\|^2|$ vs. iteration, where $(\mathbf{g}_{\mathbf{x}_t}, \mathbf{g}_{\mathbf{y}_t})$ denotes stochastic gradients.

and $\mathbf{y} \in \mathbb{R}^{d_2}$, both in high-dimensional space. The problems of interest can be categorized into non-overparametrization (NOP) and overparametrization (OP), based on whether the dimension of variables $(d_1 + d_2)$ is greater than dimension of dom $f$,

$$\textbf{NOP:} \quad \min_{\mathbf{x},\mathbf{y}} f_n(\mathbf{x}\mathbf{y}^\top) = \mathbb{E}_{\xi\sim\mathcal{D}}\big[f_n^\xi(\mathbf{x}\mathbf{y}^\top)\big], \tag{1a}$$

$$\textbf{OP:} \quad \min_{\mathbf{x},\mathbf{y}} f_o(\mathbf{x}^\top\mathbf{y}) = \mathbb{E}_{\xi\sim\mathcal{D}}\big[f_o^\xi(\mathbf{x}^\top\mathbf{y})\big]. \tag{1b}$$

Here, $d_1 = d_2$ is assumed for OP, and $\mathcal{D}$ denotes the training data. For both cases, the losses are nonconvex in $(\mathbf{x}, \mathbf{y})$. Scale-invariance refers to that $(\alpha\mathbf{x}, \mathbf{y}/\alpha)$ share the same objective value $\forall \alpha \neq 0$. It naturally calls for implicit regularization from optimization algorithms to determine the value of $\alpha$. We focus on two-variable problems in the main text for simplicity and generalize the results to multi-layer cases in the appendix. Problems (1a) and (1b) are inspired by widely-adopted modules in deep learning, where low rank adapters (LoRA) for finetuning language models is NOP, and softmax in attention falls in OP framework (Hu et al., 2022; Vaswani et al., 2017).

This work studies SAM's implicit regularization on *balancedness*, defined as $\mathcal{B}_t = \frac{1}{2}\big(\|\mathbf{x}_t\|^2 - \|\mathbf{y}_t\|^2\big)$. Balancedness is a useful alternative to sharpness for (1) because: i) it enables us to go beyond local minima and describe the behavior over SAM's entire trajectory; ii) analyses and assumptions can be significantly simplified when working with $\mathcal{B}_t$; and, iii) it enables a data-driven perspective for understanding SAM. Building on balancedness, we answer our major questions.

For Q1, we prove that even with imbalanced initialization, SAM drives $|\mathcal{B}_t| \to 0$ for OP, while ensuring a small $|\mathcal{B}_t|$ in NOP. In contrast, we also prove that balancedness of SGD is unchanged over iterations. This clear distinction between SAM and SGD is illustrated in Fig. 1. Thanks to the adoption of balancedness, our results on implicit regularization have no requirement on the batchsize compared to (Wen et al., 2023a) and can be extended to explain $m$-sharpness in (Foret et al., 2021).

Regarding Q2, we present analytical and empirical evidences that data anomalies (e.g., samples with large noise) have stronger impact on balancedness for both NOP and OP. Fig. 1 showcases an example where SAM is applied on the same problem with different SNRs. Smaller SNR (i.e., larger $\alpha$) promotes balancedness faster. Being more balanced with noisy data also aligns well with previous studies (Chen et al., 2023; Wang et al., 2023), which show that SAM performs better than SGD under data anomalies. This data-driven behavior of SAM is well depicted through balancedness.

Our theoretical understanding on balancedness also cultivates practical tools. In particular, we explicify the implicit regularization of SAM as a *data-driven* regularizer. When applied on top of, e.g., SGD, it enables a computationally efficient variant of SAM, balancedness-aware regularization (BAR), suited for scale-invariant problems such as finetuning language models with LoRA (Hu et al., 2022). BAR eliminates the need to compute the second gradient in SAM, thereby significantly reducing overhead in large-scale settings. BAR improves the test performance of LoRA on three representative downstream tasks on RoBERTa, GPT2, and OPT, while saving $95\%$ computational overhead of SAM. Moreover, this is the *first* efficient SAM approach derived from SAM's implicit regularization. In a nutshell, our contribution can be summarized as:

❖ **Theories.** Balancedness is introduced as a new metric for implicit regularization in SAM. Compared to sharpness, balancedness enables us to depict richer behaviors – SAM favors balanced solutions for both NOP and OP, and data anomalies have stronger regularization on balancedness.

❖ **Practice.** Implicit regularization of SAM is made explicit for practical merits. The resulting approach, balancedness-aware regularization (BAR), improves accuracy for finetuning language models with LoRA, while significantly saving computational overhead of SAM.

**Notation.** Bold lowercase (capital) letters denote column vectors (matrices); $\|\cdot\|$ stands for $\ell_2$ (Frobenius) norm of a vector (matrix), and $(\cdot)^\top$ refers to transpose.

## 1.1 Related Work

Related topics are streamlined here, with comprehensive discussions deferred to Apdx. A.2.

**Scale-invariance in deep learning.** Scale-invariant modules are prevalent in modern neural networks, such as LoRA, ReLU networks, and softmax in attention. However, scale-invariant problems are not yet fully understood, especially from a theoretical perspective. Neyshabur et al. (2018) develop scale-invariant PAC-Bayesian bounds for ReLU networks. A scale-invariant SGD is developed in (Neyshabur et al., 2015), and this approach becomes more practical recently in (Gonon et al., 2024). Linear neural networks entail scale-invariance and overparametrization simultaneously, and IR of (S)GD on quadratic loss is established in (Arora et al., 2018; Du et al., 2018; Gidel et al., 2019). IR of GD for softmax attention in transformers is studied in (Sheen et al., 2024) assuming linearly separable data. It is pointed out in (Dinh et al., 2017) that sharpness is sensitive to scaling, while our results indicate that when taking the training trajectory into account, SAM excludes extreme scaling.

**Mechanism behind SAM.** To theoretically explain the success of SAM, Bartlett et al. (2023) analyze sharpness on quadratic losses. Wen et al. (2023a) focus on sharpness of SAM near the solution manifold on smooth loss functions, requiring batchsize to be 1 in the stochastic case. Andriushchenko and Flammarion (2022) consider sparsity of SAM on (overparametrized) diagonal linear networks on a regression problem. Chen et al. (2023) study the benign overfitting of SAM on a two-layer ReLU network. In general, existing studies on SAM's implicit regularization focus more on sharpness and do not fully capture scale-invariance. In comparison, our results i) are Hessian-free and hence sharpness-free; ii) have no constraint on batchsize; and iii) hold for both NOP and OP.

**SAM variants.** Approaches in (Kim et al., 2022; Kwon et al., 2021) modify SAM for efficiency under coordinate-wise ill-scaling, while our results suggest that SAM favors balancedness between layers. Computationally efficient SAM variants are developed through reusing or sparsifying gradients (Liu et al., 2022; Mi et al., 2022); stochastic perturbation (Du et al., 2022a); switching to SGD (Jiang et al., 2023); and connecting with distillation (Du et al., 2022b). Our BAR can be viewed as resource-efficient SAM applied specifically for scale-invariant problems such as LoRA. Different from existing works, BAR is the first to take inspiration from the implicit regularization of SAM.

## 2 Preliminaries

This section briefly reviews SAM and then compares sharpness with balancedness. For a smoother presentation, our main numerical benchmark, LoRA (Hu et al., 2022), is revisited in Sec. 5.

### 2.1 Recap of SAM

Sharpness-aware minimization (SAM) is designed originally to seek for solutions in flat basins. The idea is formalized by enforcing small loss around the entire neighborhood in parameter space, i.e., $\min_{\mathbf{w}} \max_{\|\boldsymbol{\epsilon}\| \le \rho} h(\mathbf{w} + \boldsymbol{\epsilon})$, where $\rho$ is the radius of considered neighborhood, and $h(\mathbf{w}) := \mathbb{E}_\xi[h^\xi(\mathbf{w})]$. Practical implementation of SAM is summarized under Alg. 1. It is proved in (Wen et al., 2023a)

---

**Algorithm 1** SAM (Foret et al., 2021)

1: **Initialize:** $\mathbf{w}_0, \rho, T, \eta$
2: **for** $t = 0, \ldots, T-1$ **do**
3:     Sample $\xi$ to get a minibatch $\mathcal{M}_t$
4:     Define stochastic gradient on $\mathcal{M}_t$ as $\nabla h_t(\cdot)$
5:     Find $\boldsymbol{\epsilon}_t = \rho \nabla h_t(\mathbf{w}_t)/\|\nabla h_t(\mathbf{w}_t)\|$
6:     Update via $\mathbf{w}_{t+1} = \mathbf{w}_t - \eta \nabla h_t(\mathbf{w}_t + \boldsymbol{\epsilon}_t)$
7: **end for**

---

that $\|\nabla h_t(\mathbf{w})\| \ne 0$ (in line 5) holds for any $\rho$ under most initialization. Based on this result and similar to (Dai et al., 2023), we assume that SAM iterates are well-defined.

**Limitation of sharpness.** Coming naturally with SAM is the so-termed sharpness, given by $\mathcal{S}(\mathbf{w}) :=$ $\max_{\|\boldsymbol{\epsilon}\| \le \rho} h(\mathbf{w} + \boldsymbol{\epsilon}) - h(\mathbf{w})$. When $\|\nabla h(\mathbf{w})\| \to 0$, $\mathcal{S}(\mathbf{w})$ can be approximated using (scaled) largest eigenvalue of Hessian (Zhuang et al., 2022). This approximation is widely exploited in literature to study the implicit regularization of SAM. Consequently, most results only hold *locally* – behaviors near $\|\nabla h(\mathbf{w})\| \to 0$ are studied. In addition, sharpness (the largest eigenvalue) is not always informative for scale-invariant problems (1). Consider $h(x, y) = xy$ for example. The sharpness is 1 for any $(x, y)$ – these points are not distinguishable in terms of sharpness.

## 2.2 Prelude on Balancedness

Balancedness $\mathcal{B}_t := \frac{1}{2}(\|\mathbf{x}_t\|^2 - \|\mathbf{y}_t\|^2)$ turns out to be an intriguing alternative to sharpness on the scale-invariant problem (1). Being a global metric, balancedness is capable of describing the entire trajectory of an algorithm, regardless of proximity to critical points or definiteness of Hessian.

How does $\mathcal{B}_t$ evolve in different algorithms? To set a comparing benchmark of SAM, we first borrow results from previous works on SGD. Following implicit regularization literature such as (Arora et al., 2018, 2019b; Wen et al., 2023a), we consider SGD with infinitesimally small learning rate $\eta \to 0$ for the NOP problem (1a)

$$\mathbf{x}_{t+1} = \mathbf{x}_t - \eta \mathbf{g}_{\mathbf{x}_t}, \quad \mathbf{y}_{t+1} = \mathbf{y}_t - \eta \mathbf{g}_{\mathbf{y}_t}. \tag{2}$$

**Theorem 1** ((Arora et al., 2018, 2019a; Ji and Telgarsky, 2019; Ahn et al., 2023)). *When applying SGD on the NOP (1a), the limiting flow with $\eta \to 0$ satisfies $\|\mathbf{x}_t\|^2 - \|\mathbf{y}_t\|^2 = \|\mathbf{x}_0\|^2 - \|\mathbf{y}_0\|^2$ for all $t > 0$. In other words, $\frac{d\mathcal{B}_t}{dt} = 0$ holds.*

Theorem 1 shows that $\mathcal{B}_t \equiv \mathcal{B}_0$ given $\eta \to 0$. A graphical illustration can be found in Fig. 1 (a). Another interesting observation is that given the same initialization, $\mathcal{B}_t$ is fixed for SGD regardless of training datasets. This suggests that SGD is less adaptive to data. A similar result of Theorem 1 can be established for SGD on OP. The full statement is deferred to Apdx. C.1; see also Fig. 1 (b).

**Merits of being balance.** Because $\mathcal{B}_0$ is preserved, SGD is sensitive to initialization. For example, $(\mathbf{x}_0, \mathbf{y}_0)$ and $(2\mathbf{x}_0, 0.5\mathbf{y}_0)$ can result in extremely different trajectories, although the same objective value is shared at initialization. Most of existing works initialize $\mathcal{B}_0 \approx 0$ to promote optimization benefits, because the variance of stochastic gradient is small and the local curvature is harmonized around a balanced solution. Take the stochastic gradient of NOP on minibatch $\mathcal{M}$ for example

$$\mathbf{g}_{\mathbf{x}} = \frac{1}{|\mathcal{M}|} \Big[ \sum_{\xi \in \mathcal{M}} \nabla f_n^\xi(\mathbf{x}\mathbf{y}^\top) \Big] \mathbf{y}, \quad \mathbf{g}_{\mathbf{y}} = \frac{1}{|\mathcal{M}|} \Big[ \sum_{\xi \in \mathcal{M}} \nabla f_n^\xi(\mathbf{x}\mathbf{y}^\top) \Big]^\top \mathbf{x}. \tag{3}$$

Assuming bounded variance $\mathbb{E}[\|\frac{1}{|\mathcal{M}|} \sum_{\xi \in \mathcal{M}} \nabla f_n^\xi(\mathbf{x}\mathbf{y}^\top) - \nabla f_n(\mathbf{x}\mathbf{y}^\top)\|^2] \le \sigma^2$, it can be seen that the variance of $[\mathbf{g}_{\mathbf{x}}, \mathbf{g}_{\mathbf{y}}]$ is bounded by $\sigma^2(\|\mathbf{x}\|^2 + \|\mathbf{y}\|^2)$. In other words, among $\{(\mathbf{x}, \mathbf{y})|\mathbf{x}\mathbf{y}^\top = \mathbf{W}\}$, gradient variance is minimized if $\|\mathbf{x}\| = \|\mathbf{y}\|$, i.e., being balance. Moreover, block smoothness parameters $L_n^\mathbf{x}$ and $L_n^{\mathbf{y}}$[1] also hint upon the difficulties for optimization, where large values typically correspond to slow convergence (Bottou et al., 2018; Nesterov, 2004). With the help of Assumption 1 (in the next subsection), it can be seen that $L_n^\mathbf{x} = L_n\|\mathbf{y}\|^2$ and $L_n^{\mathbf{y}} = L_n\|\mathbf{x}\|^2$. In other words, a large $|\mathcal{B}_t|$ implies difficulty for optimizing one variable than the other. For these reasons, balancedness is well-appreciated in domains such as matrix factorization/sensing – a special case of (1a) (Tu et al., 2016; Bartlett et al., 2018; Du et al., 2018; Ge et al., 2017). It is also observed that balanced neural networks are easier to optimize relative to unbalanced ones (Neyshabur et al., 2015).

## 2.3 Assumptions and Prerequisites

To gain theoretical insights of scale-invariant problems in (1), we assume that the loss has Lipschitz continuous gradient on dom $f$ following common nonconvex optimization and SAM analyses (Bottou et al., 2018; Andriushchenko and Flammarion, 2022; Wen et al., 2023a).

**Assumption 1.** *Let $\mathbf{W} \in \mathbb{R}^{d_1 \times d_2}$, and $w \in \mathbb{R}$. For each $\xi$, $f_n^\xi(\mathbf{W})$ and $f_o^\xi(w)$ in (1) have $L_n$, and $L_o$ Lipschitz continuous gradient, respectively.*

---

[1]Definition of $L_n^\mathbf{x}$: for a fixed $\mathbf{y}$, $\|\mathbf{g}_{\mathbf{x}_1} - \mathbf{g}_{\mathbf{x}_2}\| \le L_n^\mathbf{x}\|\mathbf{x}_1 - \mathbf{x}_2\|$.

Scale-invariant problems are challenging to solve even on simple problems in Fig. 1. Even GD can diverge on some manually crafted initialization (De Sa et al., 2015; Arora et al., 2019a). With proper hyperparameters this rarely happens in practice; hence, we focus on scenarios where SGD and SAM do not diverge. This assumption is weaker than the global convergence needed in (Andriushchenko and Flammarion, 2022), and is similar to the assumption on existence (Wen et al., 2023a).

## 3 SAM for Non-Overparametrized Problems

This section tackles the implicit regularization of SAM on NOP (1a). Motivated by practical scenarios such as LoRA, we focus on cases initialized with large $|\mathcal{B}_0|$.

When ambiguity is absent, the subscript in $f_n$ and $L_n$ is ignored in this section for convenience. Applying Alg. 1 on NOP, the update of SAM can be written as

$$\tilde{\mathbf{x}}_t = \mathbf{x}_t + \rho u_t \mathbf{g}_{\mathbf{x}_t}, \quad \tilde{\mathbf{y}}_t = \mathbf{y}_t + \rho u_t \mathbf{g}_{\mathbf{y}_t} \tag{4a}$$

$$\mathbf{g}_{\tilde{\mathbf{x}}_t} = \nabla f_t(\tilde{\mathbf{x}}_t \tilde{\mathbf{y}}_t^\top) \tilde{\mathbf{y}}_t, \quad \mathbf{g}_{\tilde{\mathbf{y}}_t} = \left[ \nabla f_t(\tilde{\mathbf{x}}_t \tilde{\mathbf{y}}_t^\top) \right]^\top \tilde{\mathbf{x}}_t \tag{4b}$$

$$\mathbf{x}_{t+1} = \mathbf{x}_t - \eta \mathbf{g}_{\tilde{\mathbf{x}}_t}, \quad \mathbf{y}_{t+1} = \mathbf{y}_t - \eta \mathbf{g}_{\tilde{\mathbf{y}}_t} \tag{4c}$$

where $\rho > 0$ is the radius of SAM perturbation; $u_t := 1/\sqrt{\|\mathbf{g}_{\mathbf{x}_t}\|^2 + \|\mathbf{g}_{\mathbf{y}_t}\|^2}$; and $f_t, \nabla f_t$ denote the loss, stochastic gradient on minibatch $\mathcal{M}_t$, respectively.

**Theorem 2.** *(Dynamics of SAM.) Suppose that Assumption 1 holds. Consider SAM for NOP in* (4) *with a sufficiently small $\rho$. Let $\mathcal{B}_t := \frac{1}{2}\left(\|\mathbf{x}_t\|^2 - \|\mathbf{y}_t\|^2\right)$. For some $|\mathcal{A}_t| = \mathcal{O}(\rho^2 L)$ and $\eta \to 0$, the limiting flow of SAM guarantees that*

$$\frac{d\mathcal{B}_t}{dt} = \rho \frac{\|\mathbf{g}_{\mathbf{x}_t}\|^2 - \|\mathbf{g}_{\mathbf{y}_t}\|^2}{\sqrt{\|\mathbf{g}_{\mathbf{x}_t}\|^2 + \|\mathbf{g}_{\mathbf{y}_t}\|^2}} + \mathcal{A}_t. \tag{5}$$

*Moreover, the change on $\mathcal{B}_t$ depends on the difference of stochastic gradients on $\mathbf{x}_t$ and $\mathbf{y}_t$, i.e.,*

$$\rho \left| \|\mathbf{g}_{\mathbf{x}_t}\| - \|\mathbf{g}_{\mathbf{y}_t}\| \right| - \mathcal{O}(\rho^2 L) \leq \left| \frac{d\mathcal{B}_t}{dt} \right| \leq \rho \sqrt{\left| \|\mathbf{g}_{\mathbf{x}_t}\|^2 - \|\mathbf{g}_{\mathbf{y}_t}\|^2 \right|} + \mathcal{O}(\rho^2 L). \tag{6}$$

Unlike SGD for which $\frac{d\mathcal{B}_t}{dt} = 0$, Theorem 2 states that the balancedness for SAM is driven by gradient difference $\|\mathbf{g}_{\mathbf{x}_t}\|^2 - \|\mathbf{g}_{\mathbf{y}_t}\|^2$. To gain some intuition, if we *estimate* $\|\mathbf{g}_{\mathbf{x}_t}\|^2 - \|\mathbf{g}_{\mathbf{y}_t}\|^2 \propto \|\mathbf{y}_t\|^2 - \|\mathbf{x}_t\|^2$ based on (3) and ignore $\mathcal{A}_t$, it can be seen that $\frac{d\mathcal{B}_t}{dt} \propto -\rho \mathcal{B}_t$. This indicates the contraction on $|\mathcal{B}_t|$. A graphical illustration on decreasing $|\mathcal{B}_t|$, and its relation with gradient difference can be found in Figs. 1 (a) and 2 (a). Moreover, this implicit regularization on balancedness is global as it holds for all $t$ regardless of whether $(\mathbf{x}_t, \mathbf{y}_t)$ is close to local optima. Thanks to adopting balancedness as the metric, Theorem 2 also poses no requirement on the batchsize.

**SAM promotes balancedness.** As discussed in Section 2.2, unbalancedness is burdensome for optimization. SAM overcomes this by implicitly favoring relatively balanced solutions.

**Corollary 1.** *(Informal.) Under some regularity conditions, there exists $\bar{\mathcal{B}}_t^\rho \geq 0$ such that whenever $|\mathcal{B}_t| > \bar{\mathcal{B}}_t^\rho$, the magnitude of $\mathcal{B}_t$ shrinks, where $\bar{\mathcal{B}}_t^\rho$ can be found in (21) at appendix.*

Corollary 1 shows that SAM promotes balancedness until $|\mathcal{B}_t|$ reaches lower bounds $\bar{\mathcal{B}}_t^\rho$. Because $\bar{\mathcal{B}}_t^\rho$ depends on SAM's trajectory, we plot $\frac{1}{T} \int_0^T \bar{\mathcal{B}}_t^\rho dt$ using dotted lines for better visualization in Fig. 2 (a). It can be seen that our calculation on $\bar{\mathcal{B}}_t^\rho$ almost matches the balancedness of SAM after sufficient convergence. Being balance also reveals that the benefit of SAM can come from optimization, which is a perspective typically ignored in literature.

**Noisy data have stronger impact on balancedness.** Although our discussions extend to more general problems, for simplicity we consider the example in Fig. 2 (a), i.e., $\mathbb{E}[\|\mathbf{x}\mathbf{y}^\top - (\mathbf{A} + \alpha \mathbf{N})\|^2]$, where $\mathbf{A}$ is ground truth; $\mathbf{N}$ is data noise; and $\alpha$ determines SNR. For this problem, noisy data directly lead to noisy gradients. It can be seen in Fig. 2 (a) that smaller SNR coincides with faster decreasing of $|\mathcal{B}_t|$. To explain such a data-responsive behavior in implicit regularization, Theorem 2 states that balancedness changes largely when the difference of $\|\mathbf{g}_{\mathbf{y}_t}\|$ and $\|\mathbf{g}_{\mathbf{x}_t}\|$ is large. Since $\mathbb{E}[\|\mathbf{g}_{\mathbf{y}_t}\|^2 - \|\mathbf{g}_{\mathbf{x}_t}\|^2] \propto \alpha^2$ if assuming elements of $\mathbf{N}$ to be iid unit Gaussian variables, it thus implies that a small SNR (large $\alpha$) offers large regularization on balancedness.

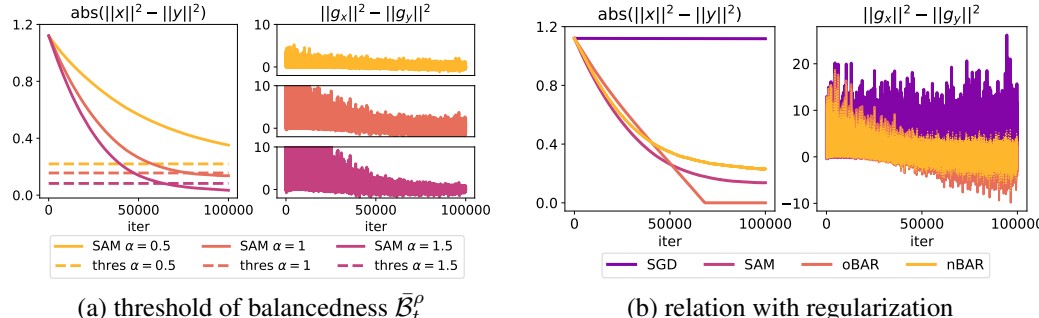

Figure 2: Implicit regularization of SAM on NOP $\mathbb{E}[\|\mathbf{x}\mathbf{y}^\top - (\mathbf{A} + \alpha\mathbf{N})\|^2]$, where $\alpha$ controls SNR. (a) the threshold of balancedness $\bar{\mathcal{B}}_t^\rho$ in Corollary 1; (b) implicit vs. explicit regularization.

**Extension to LoRA (multi-layer two-variable NOP).** For LoRA, the objective is to minimize $D$ blocks of variables simultaneously, i.e., $\min \mathbb{E}_\xi[f^\xi(\{\mathbf{x}_l\mathbf{y}_l^\top\}_{l=1}^D)]$. It is established in Theorem 5 in appendix that SAM cultivates balancedness in a layer-wise fashion, i.e., the magnitude of $\mathcal{B}_{t,l} := \frac{1}{2}(\|\mathbf{x}_{t,l}\|^2 - \|\mathbf{y}_{t,l}\|^2)$ cannot be large for each $l$. However, the $|d\mathcal{B}_{t,l}/dt|$ can be $\mathcal{O}(\sqrt{D})$ times smaller than Theorem 2 in the worst case because of the additional variables.

**Validation of IR on modern architectures.** Going beyond the infinitesimally small step size, we adopt $\eta = 0.1$ on modern language models to validate our theoretical findings. We consider finetuning a RoBERTa-large with LoRA for few-shot learning tasks. More details can be found later in Section 6.1. Balancedness of SAM and SGD on different layers in various datasets are plotted in Fig. 3. SAM has a clear trend of promoting balancedness, aligning well with our theoretical predictions.

## 4 SAM for Overparametrized Problems

Next, we focus on SAM's implicit regularization on OP (1b). Overparametrization enables SAM to have stronger regularization on balancedness. Subscripts in $f_o$ and $L_o$ are omitted for convenience. SAM's per iteration update for OP can be summarized as

$$\tilde{\mathbf{x}}_t = \mathbf{x}_t + \rho u_t \mathbf{y}_t, \quad \tilde{\mathbf{y}}_t = \mathbf{y}_t + \rho u_t \mathbf{x}_t \tag{7a}$$

$$\mathbf{g}_{\tilde{\mathbf{x}}_t} = f_t'(\tilde{\mathbf{x}}_t^\top \tilde{\mathbf{y}}_t)\tilde{\mathbf{y}}_t, \quad \mathbf{g}_{\tilde{\mathbf{y}}_t} = f_t'(\tilde{\mathbf{x}}_t^\top \tilde{\mathbf{y}}_t)\tilde{\mathbf{x}}_t \tag{7b}$$

$$\mathbf{x}_{t+1} = \mathbf{x}_t - \eta\mathbf{g}_{\tilde{\mathbf{x}}_t}, \quad \mathbf{y}_{t+1} = \mathbf{y}_t - \eta\mathbf{g}_{\tilde{\mathbf{y}}_t} \tag{7c}$$

where $u_t := \text{sgn}(f_t'(\mathbf{x}_t^\top\mathbf{y}_t))/\sqrt{\|\mathbf{x}_t\|^2 + \|\mathbf{y}_t\|^2}$; $f_t$ and $f_t'$ denote the loss, stochastic gradient on minibatch $\mathcal{M}_t$, respectively. Different from NOP, SAM has stronger regularization on balancedness, where $|\mathcal{B}_t|$ decreases whenever the norm of stochastic gradient is large. To see this, it is convenient to define $\mathcal{C}_t := |\|\mathbf{x}_t\| - \|\mathbf{y}_t\||$. Note that $\mathcal{C}_t \leq \sqrt{2|\mathcal{B}_t|}$.

**Theorem 3.** *Consider $\eta \to 0$ for (7). The limiting flow of SAM on OP ensures a decreasing magnitude of $\mathcal{B}_t$ whenever $|f_t'(\mathbf{x}_t^\top\mathbf{y}_t)| \cdot \mathcal{C}_t > \mathcal{O}(\rho L|\mathcal{B}_t|)$. Moreover, the speed of decrease can be lower- and upper- bounded as*

$$\rho|f_t'(\mathbf{x}_t^\top\mathbf{y}_t)| \cdot \mathcal{C}_t - \mathcal{O}(\rho^2 L|\mathcal{B}_t|) \leq \left|\frac{d\mathcal{B}_t}{dt}\right| \leq \rho|f_t'(\mathbf{x}_t^\top\mathbf{y}_t)|\sqrt{2|\mathcal{B}_t|} + \mathcal{O}(\rho^2 L|\mathcal{B}_t|).$$

Given $\rho \to 0$ and sufficiently noisy data, Theorem 3 implies that $|\mathcal{B}_t| \to 0$. Moreover, Theorem 3 also states that the regularization power on balancedness is related to both gradient norm and balancedness itself. The elbow-shaped curve of $|\mathcal{B}_t|$ in Fig. 1 (b) demonstrates that the regularization power is reducing, as both gradient norm and balancedness shrink over time.

**Noisy data have stronger impact on balancedness.** As shown in Fig. 1 (b), balancedness is promoted faster on problems with lower SNR. This data-responsive behavior can be already seen from Theorem 3, because $|d\mathcal{B}_t/dt|$ is directly related with $|f_t'(\mathbf{x}_t^\top\mathbf{y}_t)|$, and $\mathbb{E}[|f_t'(\mathbf{x}_t^\top\mathbf{y}_t)|]$ is clearly larger when data are more noisy. In other words, SAM exploits noisy data for possible optimization merits from balancedness (see discussions in Sec. 2.2). Overall, the implicit regularization on balancedness aligns well with the empirical observations in presence of data anomalies (Wang et al., 2023; Sherborne et al., 2023), where SAM outperforms SGD by a large margin.

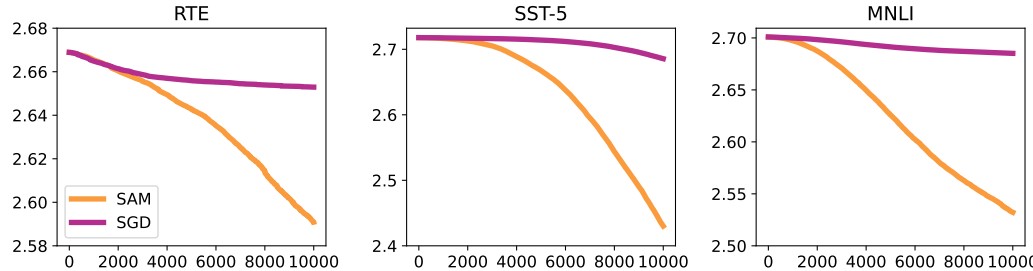

Figure 3: Implicit regularization of SAM on LoRA. We consider few shot learning with LoRA on a RoBERTa-large. For datasets RTE, SST-5, and MNLI, 1st, 12th and 24th query layers' $2|\mathcal{B}_{t,l}|$ are plotted, respectively. The layers are chosen to represent early, middle, and final stages of RoBERTa. The averaged $\bar{\mathcal{B}}_{t,l}^{\rho}$ in Corollary 1 is 0.37, 0.21, and 0.29, respectively.

**Extension to $m$-sharpness.** $m$-sharpness is a variant of SAM suitable for distributed training. It is observed to empirically improve SAM's performance (Foret et al., 2021). $m$-sharpness evenly divides minibatch $\mathcal{M}_t$ into $m$ disjoint subsets, i.e., $\{f_{t,j}\}_{j=1}^m$, and perform SAM update independently on each subset; see (38) in appendix. It turns out that $m$-sharpness can also be explained using balancedness. With formal proofs in Apdx. C.3, the IR of $m$-sharpness amounts to substitute $|f_t'(\mathbf{x}_t^\top \mathbf{y}_t)|$ in Theorem 3 with $\frac{1}{m}\sum_{j=1}^m |f_{t,j}'(\mathbf{x}_t^\top \mathbf{y}_t)|$. This means that the regularization on balancedness from $m$-sharpness is more profound than vanilla SAM, because $\frac{1}{m}\sum_{j=1}^m |f_{t,j}'(\mathbf{x}_t^\top \mathbf{y}_t)| \geq |f_t'(\mathbf{x}_t^\top \mathbf{y}_t)|$.

Finally, we connect balancedness with sharpness on local minima of OP.

**Lemma 1.** *Let $\mathcal{W}^* = \{(\mathbf{x}, \mathbf{y})|\mathbf{x}^\top \mathbf{y} = w, f'(w) = 0, f''(w) > 0\}$ be non-empty. For the OP problem (1b), minimizing sharpness within $\mathcal{W}^*$ is equivalent to finding $\mathcal{B} = 0$ in $\mathcal{W}^*$.*

This link showcases that by studying balancedness we can also obtain the implicit regularization on sharpness for free. A concurrent work also links balancedness with sharpness (the largest eigenvalue) for some one-hidden layer neural networks (Singh and Hofmann, 2024). Compared with (Wen et al., 2023a), this is achieved with less assumptions and simplified analyses. More importantly, balancedness enables us to cope with arbitrary batchsize, to explain SAM's stronger regularization with noisy data, and to extend results to $m$-sharpness.

## 5 Implicit Regularization Made Explicit

Next, insights from our theoretical understanding of SAM are leveraged to build practical tools. We adopt LoRA (Hu et al., 2022) as our major numerical benchmark for scale-invariant problems given its prevalence in practice. More diverse examples on both OP and NOP can be found in Apdx. A.3. Compared to full parameter-tuning, LoRA is more economical in terms of memory not only for finetuning, but also for serving multiple downstream tasks. LoRA and its variants are actively developed and well welcomed by the community; see e.g., HuggingFace's PEFT codebase.[2]

### 5.1 Overview of LoRA

Given a pretrained model with frozen weight $\mathbf{W}_l \in \mathbb{R}^{d_1 \times d_2}$ on a particular layer $l$, the objective of LoRA is to find low rank matrices $\mathbf{X}_l \in \mathbb{R}^{d_1 \times r}$, and $\mathbf{Y}_l \in \mathbb{R}^{d_2 \times r}$ with $r \ll \min\{d_1, d_2\}$ such that the loss is minimized for a downstream task, i.e.,

$$\min_{\{\mathbf{X}_l, \mathbf{Y}_l\}_l} \mathcal{L}\big(\{\mathbf{W}_l + \mathbf{X}_l \mathbf{Y}_l^\top\}_l\big). \tag{8}$$

LoRA enjoys parameter efficiency for finetuning thanks to the low-rank matrices $\mathbf{X}_l$ and $\mathbf{Y}_l$. For instance, it only requires 0.8M trainable parameters to finetune a 355M-parameter RoBERTa-large (Hu et al., 2022). The outer product of $\mathbf{X}_l$ and $\mathbf{Y}_l$ induces scale-invariance, and the number of variables renders it NOP. The downside of LoRA, on the other hand, is the drop on test performance due to the parsimony on trainable parameters. Unbalancedness is also unavoidable for LoRA, due

---

[2]https://github.com/huggingface/peft

| **Algorithm 2** nBAR | **Algorithm 3** oBAR |
|---|---|
| 1: **Initialize:** learning rate $\{\eta_t\}$, regularization coefficient $\{\alpha_t\}$ | 1: **Initialize:** learning rate $\{\eta_t\}$, regularization coefficient $\{\alpha_t\}$ |
| 2: **for** $t = 0, \ldots, T - 1$ **do** | 2: **for** $t = 0, \ldots, T - 1$ **do** |
| 3:      Get stochastic gradient $\mathbf{g}_{\mathbf{x}_t}$ and $\mathbf{g}_{\mathbf{y}_t}$ | 3:      Get stochastic gradient $\mathbf{g}_{\mathbf{x}_t}$ and $\mathbf{g}_{\mathbf{y}_t}$ |
| 4:      **if** $\|\mathbf{g}_{\mathbf{x}_t}\| \geq \|\mathbf{g}_{\mathbf{y}_t}\|$ **then** | 4:      **if** $\|\mathbf{x}_t\| \geq \|\mathbf{y}_t\|$ **then** |
| 5:         $\mathbf{x}_t \leftarrow (1 + \alpha_t \eta_t)\mathbf{x}_t$ | 5:         $\mathbf{x}_t \leftarrow (1 - \alpha_t \eta_t)\mathbf{x}_t$ |
| 6:         $\mathbf{y}_t \leftarrow (1 - \alpha_t \eta_t)\mathbf{y}_t$ | 6:         $\mathbf{y}_t \leftarrow (1 + \alpha_t \eta_t)\mathbf{y}_t$ |
| 7:      **else** | 7:      **else** |
| 8:         $\mathbf{x}_t \leftarrow (1 - \alpha_t \eta_t)\mathbf{x}_t$ | 8:         $\mathbf{x}_t \leftarrow (1 + \alpha_t \eta_t)\mathbf{x}_t$ |
| 9:         $\mathbf{y}_t \leftarrow (1 + \alpha_t \eta_t)\mathbf{y}_t$ | 9:         $\mathbf{y}_t \leftarrow (1 - \alpha_t \eta_t)\mathbf{y}_t$ |
| 10:      **end if** | 10:      **end if** |
| 11:      Optimizer update (via Adam or SGD) | 11:      Optimizer update (via Adam or SGD) |
| 12: **end for** | 12: **end for** |

to the need of initializing at $\mathbf{X}_l \sim \mathcal{N}(0, \sigma^2), \mathbf{Y}_l = \mathbf{0}$; see an example of RoBERTa-large in Fig. 3. The unbalancedness leads to instability of LoRA when finetuning RoBERTa on datasets SST-2 and MNLI; see more details in Apdx. D.4.

Integrating SAM with LoRA is a case with mutual benefits – LoRA reduces the additional memory requirement of SAM, while SAM not only overcomes the distributional shift in finetuning (Zhou et al., 2022), but also mitigates the possible inefficiency associated with LoRA's unbalancedness.

## 5.2 Balancedness-Aware Regularization (BAR)

However, directly applying SAM variants on LoRA exhibits two concerns: i) SAM doubles computational cost due to the need of two gradients; and ii) additional efforts are required to integrate SAM with gradient accumulation and low-precision training (HuggingFace), which are common techniques for memory and runtime efficiency in large-scale finetuning. Note that concern i) is annoying given the size of language models, especially in setups involving model parallelism.

Our balancedness-aware regularization (BAR) is a highly efficient approach to address both concerns, and it fixes the accuracy drop of LoRA relative to full-parameter finetuning. BAR is also the *first* efficient SAM variant derived from implicit regularization. The key observation for our algorithm design is that SAM's implicit regularization on balancedness can be achieved with an explicit regularizer $\alpha_t |\mathbf{x}^\top \mathbf{x} - \mathbf{y}^\top \mathbf{y}|$. This regularizer originates from matrix sensing; see e.g., (Tu et al., 2016; Ge et al., 2017). For OP, choosing $\alpha_t := \mathcal{O}(|f'(\mathbf{x}_t^\top \mathbf{y}_t)| / \sqrt{\|\mathbf{x}_t\|^2 + \|\mathbf{y}_t\|^2})$ recovers SAM's dynamic on $\mathcal{B}_t$ up to an error of $\mathcal{O}(\rho^2)$; cf. Lemma 2 in appendix. By ignoring this error, it can be seen that $\mathcal{B}_t$ decreases when $\|\mathbf{x}_t\| \geq \|\mathbf{y}_t\|$. Following this dynamic, we regulate balancedness based on whether $\|\mathbf{x}_t\| \geq \|\mathbf{y}_t\|$. The resultant approach is termed as overparamterized BAR (oBAR) to reflect its source in OP.

On the other hand, because LoRA is NOP inherently, we take inspiration from Theorem 2 – dropping the term $\mathcal{A}_t$ and mimicking dynamics of SAM. In particular, we regulate the objective with $\alpha_t(\mathbf{x}^\top \mathbf{x} - \mathbf{y}^\top \mathbf{y})$ if $\|\mathbf{g}_{\mathbf{x}_t}\|^2 < \|\mathbf{g}_{\mathbf{y}_t}\|^2$; otherwise $\alpha_t(\mathbf{y}^\top \mathbf{y} - \mathbf{x}^\top \mathbf{x})$. The resultant approach is termed as nBAR. A graphical illustration can be found in Fig. 2 (b). It can be observed that nBAR shares similar performance as SAM on NOP. Both nBAR and oBAR can be implemented in the same manner as weight decay, and their detailed steps are summarized in Algs. 2 and 3, respectively.

Another benefit of BAR, in addition to the lightweight computation, is that it can be applied individually on each LoRA layer. As previously discussed (cf. Theorem 5), the number of layers has a negative impact on balancedness. By overcoming this "curse of multi-layer", BAR can induce better test performance over SAM.

**Schedule of $\alpha_t$.** In both nBAR and oBAR, one can employ a decreasing scheduler for $\alpha_t$ for algorithmic flexibility. This is motivated by the fact that for both NOP and OP problems, the implicit regularization of SAM is less powerful after sufficient balancedness or near optimal. Commonly adopted cosine and linear schedules work smoothly.

Table 1: Few shot learning on RoBERTa (355M). † denotes results reported by (Malladi et al., 2023)

| RoBERTa | SST-2 | SST-5 | SNLI | MNLI | RTE | TREC | avg (↑) |
|---|---|---|---|---|---|---|---|
| LoRA | $91.1_{\pm0.8}$ | $52.3_{\pm2.9}$ | $84.3\pm0.3$ | $78.1_{\pm1.3}$ | $77.5_{\pm2.3}$ | $96.6_{\pm1.0}$ | 80.0 |
| LoRA-SAM | $\mathbf{92.2}_{\pm0.4}$ | $54.2_{\pm2.0}$ | $\mathbf{85.5}_{\pm0.7}$ | $\mathbf{78.7}_{\pm1.0}$ | $\underline{80.6}_{\pm4.3}$ | $96.7_{\pm0.2}$ | **81.3** |
| **LoRA-oBAR** | $\underline{91.5}_{\pm0.9}$ | $\underline{54.5}_{\pm2.7}$ | $\underline{84.9}_{\pm0.5}$ | $78.3_{\pm2.2}$ | $79.7_{\pm2.0}$ | $96.7_{\pm0.5}$ | 80.9 |
| **LoRA-nBAR** | $91.4_{\pm0.5}$ | $\mathbf{55.0}_{\pm2.0}$ | $84.9_{\pm1.4}$ | $78.1_{\pm0.2}$ | $\mathbf{81.0}_{\pm1.0}$ | $96.7_{\pm1.0}$ | $\underline{81.2}$ |
| Zero-Shot† | 79.0 | 35.5 | 50.2 | 48.8 | 51.4 | 32.0 | 49.5 |

Table 2: Runtime of BAR (normalized to LoRA, 1x) on OPT-1.3B. SAM relies on FP32 for stability. LoRA and BAR adopt FP16 training since this is the default choice for large models. nBAR and oBAR share similar runtime, hence reported together.

| runtime (↓) | SST-2 | CB | RTE | COPA | ReCoRD | SQuAD |
|---|---|---|---|---|---|---|
| LoRA-SAM | 4.43x | 3.34x | 4.10x | 3.28x | 4.35x | 3.54x |
| **LoRA-BAR** | **1.05**x | **1.03**x | **1.04**x | **1.05**x | **1.04**x | **1.03**x |

# 6 Numerical Experiments

To demonstrate the effectiveness of BAR, numerical experiments are conducted on various deep learning tasks using language models (LMs). Bold and underlined numbers are used to highlight the best and second best performance, respectively. More experimental details can be found in Apdx. D. Code is available at `https://github.com/BingcongLi/BAR`.

## 6.1 Few-shot Learning with RoBERTa-large and OPT-1.3B

The first task to consider is few-shot learning with LoRA (Malladi et al., 2023), where the goal is to finetune a language model with a small training set. We follow the settings in (Malladi et al., 2023), and choose the backbones as RoBERTa-large, a masked LM with 355M parameters, and OPT-1.3B, an autoregressive LM (Liu et al., 2019; Zhang et al., 2022).

Results of the proposed oBAR and nBAR on RoBERTa-large are summarized in Table 1. As indicated by the zero-shot performance, the distributional shift between finetuning and pretraining datasets is obvious. This is a natural setting suitable for SAM and BAR. The averaged test accuracy is improved by 0.9 and 1.2 via oBAR and nBAR, respectively. The performance of nBAR is close to SAM. Moreover, BAR saves 74% additional runtime of SAM; see more details in Table 7 in the appendix.

The proposed nBAR and oBAR perform even better when scaling up to OPT-1.3B. BAR reduces the overhead of SAM by more than 95% because of its compatibility with FP16 training; see Table 2. Note that applying FP16 directly with SAM leads to underflow; see more in Apdx. D. This signifies the flexibility of BAR over SAM when scaling to large problems, as FP16 is the default choice for LMs. Prefix tuning (Li and Liang, 2021) is also included as a benchmark for comparisons on test performance. We report F1 score for SQuAD and accuracy for other datasets in Table 3. The averaged improvement over LoRA is 0.9 and 1.6 from oBAR and nBAR, respectively, both outperforming SAM. We conjecture that the performance gap between SAM and BAR comes from their different effectiveness in regularizing balancedness. Balancedness of a particular layer is decreasing slower in SAM due to multiple layers, as shown in Theorem 5, while BAR promotes balancedness faster as it can be applied individually on each LoRA layer. Comparing the absolute improvement for RoBERTa-large (355M) and OPT-1.3B, it is conjectured that BAR has more potential for larger models, and the verification is left for future due to hardware constraints.

## 6.2 Finetuning with RoBERTa-large

Having demonstrated the power of BAR in few-shot learning, we then apply it to finetune RoBERTa-large with LoRA. The results can be found in Table 4. It can be observed that nBAR and oBAR improve the performance of LoRA and prefix tuning (Li and Liang, 2021) on most of tested datasets.

Table 3: Performance of BAR for few shot learning using OPT-1.3B.

| OPT-1.3B | SST-2 | CB | RTE | COPA | ReCoRD | SQuAD | avg ($\uparrow$) |
|---|---|---|---|---|---|---|---|
| Prefix | $92.9_{\pm1.0}$ | $71.6_{\pm3.0}$ | $65.2_{\pm2.6}$ | $73.0_{\pm1.0}$ | $69.7_{\pm1.0}$ | $82.1_{\pm1.4}$ | 75.8 |
| LoRA | $93.1_{\pm0.2}$ | $72.6_{\pm3.7}$ | $69.1_{\pm4.8}$ | $\mathbf{78.0}_{\pm0.0}$ | $70.8_{\pm1.0}$ | $81.9_{\pm1.8}$ | 77.6 |
| LoRA-SAM | $93.5_{\pm0.5}$ | $74.3_{\pm1.0}$ | $\mathbf{70.6}_{\pm2.7}$ | $\mathbf{78.0}_{\pm0.0}$ | $\underline{70.9}_{\pm1.2}$ | $\mathbf{83.0}_{\pm0.7}$ | 78.4 |
| **LoRA-oBAR** | $\underline{93.6}_{\pm0.6}$ | $\underline{75.6}_{\pm4.5}$ | $70.4_{\pm4.8}$ | $\mathbf{78.0}_{\pm0.0}$ | $\underline{70.9}_{\pm0.8}$ | $\underline{82.5}_{\pm0.5}$ | $\underline{78.5}$ |
| **LoRA-nBAR** | $\mathbf{93.7}_{\pm0.7}$ | $\mathbf{79.8}_{\pm4.4}$ | $\underline{70.5}_{\pm2.4}$ | $\mathbf{78.0}_{\pm0.0}$ | $\mathbf{71.0}_{\pm1.0}$ | $82.3_{\pm1.8}$ | $\mathbf{79.2}$ |
| Zero-Shot | 53.6 | 39.3 | 53.1 | 75.0 | 70.2 | 27.2 | 53.1 |

Table 4: Finetuning RoBERTa (355M) with BAR. Results marked with † are taken from (Hu et al., 2022), and those with ∗ refer to Adapter[P] in (Hu et al., 2022).

| RoBERTa | # para | STS-B | RTE | MRPC | CoLA | QQP | avg ($\uparrow$) |
|---|---|---|---|---|---|---|---|
| FT[†] | 355M | 92.4 | 86.6 | 90.9 | 68.0 | 90.2 | 85.6 |
| Adapter∗ | 0.8M | 91.9±0.4 | 80.1±2.9 | 89.7±1.2 | $\mathbf{67.8}$±2.5 | $\mathbf{91.7}$±0.2 | 84.2 |
| LoRA | 0.8M | 92.4±0.1 | 88.2±0.6 | 89.6±0.5 | 64.8±1.4 | 91.4±0.1 | 85.3 |
| **LoRA-oBAR** | 0.8M | $\mathbf{92.6}$±0.1 | $\underline{88.7}$±0.2 | $\mathbf{90.3}$±0.9 | 65.1±1.0 | $\underline{91.6}$±0.1 | $\underline{85.7}$ |
| **LoRA-nBAR** | 0.8M | $\mathbf{92.6}$±0.2 | $\mathbf{89.2}$±1.3 | $\mathbf{90.3}$±0.4 | $\underline{65.6}$±1.2 | $\underline{91.6}$±0.1 | $\mathbf{85.9}$ |

Table 5: Finetuning GPT2 (345M) with BAR on WebNLG. Results of prefix tuning and full-parameter finetuning are obtained from (Hu et al., 2022).

| GPT2 | FT∗ | Prefix∗ | LoRA | **LoRA-oBAR** | **LoRA-nBAR** |
|---|---|---|---|---|---|
| # param | 354M | 0.35M | 0.35M | 0.35M | 0.35M |
| BLEU ($\uparrow$) | 46.5 | 55.1 | 54.99±0.24 | $\underline{55.15}$±0.19 | $\mathbf{55.20}$±0.16 |

On average, oBAR leads to a gain of $0.4$, and nBAR raises the test performance by $0.6$. BAR thereby fills the gap of test performance between LoRA (0.8M) and full-parameter (355M) finetuning.

### 6.3 Text Generation on GPT2-medium

Lastly, we consider BAR on a text-generation problem using GPT2-medium, a model with 345M parameters. Results on WebNLG (Gardent et al., 2017) are reported in Table 5. It can be seen that oBAR matches the performance of prefix tuning, while nBAR achieves the best BLEU score.

## 7 Discussions

This work provides theoretical and empirical evidence on the implicit regularization of SAM for both scale-invariant NOP and OP problems. Balancedness, as an alternative to commonly adopted sharpness, is employed as the metric to capture global and data-responsive behaviors of SAM. We find that i) SAM promotes variables to have (relatively) balanced norms; and ii) noisy data have stronger impact on balancedness. Lastly, we explicify the implicit regularization as a data-driven regularizer to foster the design of a computationally efficient SAM variant, termed BAR. The effectiveness of BAR is demonstrated using various tasks on RoBERTa-large, GPT2 and OPT. BAR saves $95\%$ overhead of SAM and enhances the accuracy of LoRA to the level of full-parameter finetuning.

**Limitation and Future directions.** Our approach, BAR, is best applied on scale-invariant modules in neural networks. Finetuning language models with LoRA, as a popular option in practice, is a setting naturally suitable for our approach. However, our approach does not apply for linear models, e.g., logistic regression. Regarding future directions, an interesting one is whether SAM has other forms of implicit regularization beyond balancedness and sharpness. The exploration of other scale-invariant architectures beyond LoRA, e.g., the softmax function in attention, is also deferred to future work.

## Acknowledgements

We thank anonymous reviewers for their suggestions. BL is supported by Swiss National Science Foundation (SNSF) Project Funding No. 200021-207343. LZ gratefully acknowledges funding by the Max Planck ETH Center for Learning Systems (CLS). NH is supported by ETH research grant funded through ETH Zurich Foundations and SNSF Project Funding No. 200021-207343.

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

# Supplementary Document for
# "Implicit Regularization of Sharpness-Aware Minimization for Scale-Invariant Problems"

## A  Missing Details

### A.1  Broad Impact

The theories and approaches are applicable across various scenarios. The proposed algorithmic tool simplifies finetuning language models, improves performance of downstream tasks, and consumes less resource compared to SAM. For tasks such as sentiment classification, our approach facilitates real world systems such as recommendation by improving accuracy. However, caution is advised when the downstream tasks of language models involve generation. For these tasks, users should thoroughly review generated content and consider to implement gating methods to ensure safety and trustworthiness.

### A.2  More on Related Work

**Sharpness and generalization.** Sharpness is observed to relate with generalization of SGD in deep learning (Keskar et al., 2016). It is found that sharpness varies with the ratio between learning rate and batchsize in SGD (Jastrzębski et al., 2017). Large scale experiments also indicate sharpness-based measures align with generalization in practical scenarios (Jiang et al., 2020; Chen et al., 2022). Theoretical understandings on generalization error using sharpness-related metrics can be found in e.g., (Dziugaite and Roy, 2017; Neyshabur et al., 2017; Wang and Mao, 2022). There is a large body of literature exploring sharpness for improved generalization. Entropy SGD leverages local entropy in search of a flat valley (Chaudhari et al., 2017). A similar approach as SAM is also developed in (Wu et al., 2020) while putting more emphases on adversarial robustness. Stochastic weight averaging is proposed for finding flatter minima in (Izmailov et al., 2018). It is shown later in (Wen et al., 2023b) that the interplay between sharpness and generalization subtly depends on data distributions and model architectures, and there are unveiled reasons beyond sharpness for the benefit of SAM.

**SAM variants.** Although SAM is successful in various deep learning tasks, it can be improved further by leveraging local geometry in a fine-grained manner. For example, results in (Zhao et al., 2022; Barrett and Dherin, 2021) link SAM with gradient norm penalization. Zhuang et al. (2022) optimize sharpness gap and training loss jointly. A more accurate manner to solve inner maximization in SAM is developed in (Li and Giannakis, 2023). SAM and its variants are also widely applied to domain generalization problems; see e.g., (Zhang et al., 2023b; Wang et al., 2023).

**Other perspectives for SAM.** The convergence of SAM is comprehensively studied in (Si and Yun, 2023). Agarwala and Dauphin (2023) focus on the edge-of-stability-like behavior of unnormalized SAM on quadratic problems. Dai et al. (2023) argue that the normalization in SAM, i.e., line 5 of Alg. 1, is critical. Sharpness measure is generalized to any functions of Hessian in (Tahmasebi et al., 2024). However, even the generalized sharpness cannot provide implicit regularization for simple functions such as $h(x, y) = xy$, because the Hessian is the same for all $(x, y)$. In addition, when Hessian is negative definite, some of the generalized sharpness measures (e.g., determinate of Hessian) may not be necessarily meaningful.

**Implicit regularization.** The regularization effect can come from optimization algorithms rather than directly from the regularizer in objective functions. This type of the behavior is termed as implicit regularization or implicit bias of the optimizer. The implicit regularization of (S)GD is studied from multiple perspectives, such as margin (Ji and Telgarsky, 2019; Lyu and Li, 2020), kernel (Arora et al., 2019c), and Hessian (Li et al., 2022; Arora et al., 2022). Initialization can also determine the implicit regularization (Woodworth et al., 2020). Most of these works explore the overparametrization regime.

**LoRA and parameter-efficient finetuning.** LoRA (Hu et al., 2022), our major numerical benchmark, is an instance of parameter-efficient finetuning (PEFT) approaches. PEFT reduces the resource requirement for large language models on various downstream tasks, at the cost of possible accuracy drops on test performance. The latter, together with the transfer learning setup jointly motivate the adoption of SAM. Other commonly adopted PEFT methods include, e.g., adapters (Houlsby et al., 2019) and prefix tuning (Li and Liang, 2021). There are also various efforts to further improve

LoRA via adaptivity (Zhang et al., 2023a), chaining (Xia et al., 2024), aggressive parameter saving (Kopiczko et al., 2024), low-bit training (Dettmers et al., 2023), and modifications for long-sequences (Chen et al., 2024). Most of these efforts are orthogonal to BAR proposed in this work.

### A.3 Additional Applications of Scale-Invariant Problems in Deep Learning

**Attention in transformers.** Attention is one of the backbones of modern neural networks (Vaswani et al., 2017). Given the input $\mathbf{D}$, attention can be written as

$$\min_{\mathbf{Q},\mathbf{K},\mathbf{V}} \; \text{softmax}\left(\frac{1}{\alpha}\mathbf{D}\mathbf{Q}\mathbf{K}^\top\mathbf{D}^\top\right)\mathbf{D}\mathbf{V} \tag{9}$$

where $\{\mathbf{Q}, \mathbf{K}, \mathbf{V}\}$ are query, key, and value matrices to be optimized. This is a scale-invariant problem because scaling $\{\mathbf{Q}, \mathbf{K}\}$ does not modify the objective function. Considering the number of variables, the optimization of $\{\mathbf{Q}, \mathbf{K}\}$ is considered as OP.

**Two-layer linear neural networks.** This problem is a simplified version of two-layer ReLU neural nets, and its objective can be defined as

$$f(\mathbf{W}_1, \mathbf{W}_2) = \frac{1}{2}\mathbb{E}_{(\mathbf{a},\mathbf{b})}\big[\|\mathbf{W}_1\mathbf{W}_2\mathbf{a} - \mathbf{b}\|^2\big]. \tag{10}$$

This is usually adopted as an example for overparametrization, and can be extended to deeper linear neural networks; see e.g., (Arora et al., 2019a). Moreover, it is known that the optimization for such problem is quite challenging, and GD can fail to converge if $\mathbf{W}_1$ and $\mathbf{W}_2$ are not initialized with balancedness (Arora et al., 2019a). An extension of (10) is two-layer ReLU networks, which are widely adopted in theoretical frameworks to understand the behavior of neural networks. ReLU networks are scale-invariant, but only when the scaling factor is positive.

**Other examples.** For ResNets, two-variable scale-invariant submodules also include affine Batch-Norm and the subsequent convolutional layer. For transformers, scale-invariant submodules besides attention include LayerNorm and its subsequent linear layer.

### A.4 SAM Pays More Attention to Difficult Examples

**Testing example for NOP.** *The problem presented below is adopted in Fig. 1 (a) and Fig. 2 for visualization of SAM's behavior on NOP.* We consider a special case of problem (1a), where the goal is to fit (rank-1) matrices by minimizing

$$f_n(\mathbf{x}, \mathbf{y}) = \mathbb{E}_\xi\big[\|\mathbf{x}\mathbf{y}^\top - (\mathbf{A} + \alpha\mathbf{N}_\xi)\|^2\big] \tag{11}$$

where $\mathbf{A} \in \mathbb{R}^{3\times3} := \text{diag}[0.5, 0, 0]$ and $\mathbf{N}_\xi \in \mathbb{R}^{3\times3}$ denote the ground truth and Gaussian noise, respectively; and $\alpha$ controls the SNR. Here we choose $\mathbf{N}_\xi := \text{diag}[1.0, 0.8, 0.5]\mathbf{U}_\xi$, where entries of $\mathbf{U}_\xi$ are unit Gaussian random variables.

In our simulation of Fig. 1 (a), we set the step size to be $\eta = 10^{-4}$ and the total number of iterations as $T = 10^5$ for both SGD and SAM. Parameter $\rho$ is chosen as $0.1$ for SAM. For both algorithms, initialization is $\mathbf{x}_0 = [0.2, -0.1, 0.3]^\top$ and $\mathbf{y}_0 = -3\mathbf{x}_0$. Note that we choose a small step size to mimic the settings of our theorems.

**Testing example for OP.** *The problem presented below is adopted in Fig. 1 (b) for visualization of SAM on OP.* A special case of problem (1b) is considered with objective function

$$f_o(\mathbf{x}, \mathbf{y}) = \mathbb{E}_\xi\big[\|\mathbf{x}^\top\mathbf{y} - (a + \alpha n_\xi)\|^2\big] \tag{12}$$

where $a \in \mathbb{R}$ and $n_\xi \in \mathbb{R}$ denote the ground truth and Gaussian noise, respectively. We choose $a = 0.5$ and $n_\xi$ as a unit Gaussian random variable. Here, $\alpha$ controls the SNR of this problem.

In our simulation of Fig. 1 (b), we set $\eta = 10^{-4}$ and $T = 10^5$ for both SGD and SAM. Parameter $\rho$ is set as $0.2$ for SAM. For both algorithms, initialization is $\mathbf{x}_0 = [0.2, -0.1, 0.3]^\top$ and $\mathbf{y}_0 = -3\mathbf{x}_0$.

### A.5 Scale-Invariance in OP

Scale-invariance also bothers OP in the same fashion as it burdens NOP. For completeness, the scale-invariance of OP can be verified by

$$f_o(\mathbf{x}^\top\mathbf{y}) = f_o\Big((\alpha\mathbf{x})^\top(\frac{1}{\alpha}\mathbf{y})\Big), \forall \alpha \neq 0. \tag{13}$$

An optimizer has to determine $\alpha$ for OP despite it does not influence objective value. Hence, scaling is redundant for OP.

Similar to NOP, the (stochastic) gradient of OP is not scale-invariant. In particular, given a minibatch of data $\mathcal{M}$, the stochastic gradient for OP (1b) can be written as

$$\mathbf{g_x} = \frac{1}{|\mathcal{M}|}\Big[\sum_{\xi \in \mathcal{M}} (f_o^\xi)'(\mathbf{x}^\top \mathbf{y})\Big]\mathbf{y}, \quad \mathbf{g_y} = \frac{1}{|\mathcal{M}|}\Big[\sum_{\xi \in \mathcal{M}} (f_o^\xi)'(\mathbf{x}^\top \mathbf{y})\Big]\mathbf{x}. \tag{14}$$

Consequently, being balance also brings optimization benefits for OP as discussed previously in Section 2.2 .

### A.6 BAR in Detail

BAR is inspired jointly from the balancedness-promoting regularizer $|\|\mathbf{x}_t\|^2 - \|\mathbf{y}_t\|^2|$ and the dynamics of SAM on both NOP and OP. The implementation of BAR is similar as weight decay in AdamW (Loshchilov and Hutter, 2019).

Here we use nBAR as an example. If ignoring $\mathcal{A}_t$ in Theorem 2, it can be seen that $\mathcal{B}_t$ for NOP decreases whenever $\|\mathbf{g}_{\mathbf{x}_t}\| < \|\mathbf{g}_{\mathbf{y}_t}\|$. In other words, the balancedness of SAM is driven by the difference between the gradient norms at $\mathbf{x}_t$ and $\mathbf{y}_t$. nBAR mimics this and triggers balancedness when stochastic gradients $\mathbf{g}_{\mathbf{x}_t}$ and $\mathbf{g}_{\mathbf{y}_t}$ are not balanced; see Alg. 2.

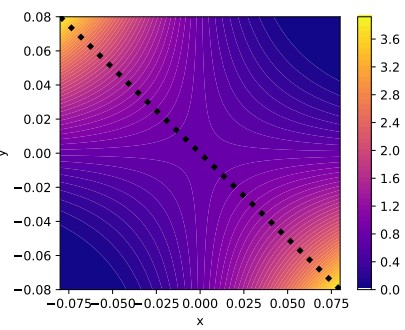

Figure 4: The value of $f(x, y)$. Once SGD reaches the dotted line, i.e., the hard constraint $|x| = |y|$, it can only converge to a saddle point $(0, 0)$.

Finally, we illustrate more on the reasons for employing regularization in OP rather than posing $\|\mathbf{x}_t\| = \|\mathbf{y}_t\|$ as a hard constraint or initializing in a balanced manner, i.e., $\|\mathbf{x}_0\| = \|\mathbf{y}_0\|$. First, it is quite clear that $\|\mathbf{x}\| = \|\mathbf{y}\|$ is a nonconvex set and how to project on such a set is still debatable. Second, the 'symmetry' associated with the scale-invariant problems does not always favor this constraint. For the purpose of graphical illustration, we consider a 2-dimensional example $f(x, y) = 30000(xy - 0.005)^2$. It is quite clear that the objective is symmetric regarding the line $x = -y$, which satisfies $|x| = |y|$; see Fig. 4. However, it is not hard to see that SGD can never leave $x = -y$ once it reaches this line via a hard constraint or initialized on this line. In other words, directly adding $\|\mathbf{x}\| = \|\mathbf{y}\|$ as a constraint can trap the algorithm at saddle points. This symmetric pattern is even more complicated in high dimension, i.e., symmetry over multiple lines or hyperplanes. Hence, one should be extremely careful about this hard constraint, and regularization is a safer and more practical choice.

## B Missing Proofs for NOP

### B.1 Proof of Theorem 1

*Proof.* For notational convenience, we let $\mathbf{G}_t := \nabla f_t(\mathbf{x}_t \mathbf{y}_t^\top)$. Then, we have that

$$\frac{d\|\mathbf{x}_t\|^2}{dt} = 2\mathbf{x}_t^\top \frac{d\mathbf{x}_t}{dt} = -2\mathbf{x}_t^\top \mathbf{g}_{\mathbf{x}_t} = -2\mathbf{x}_t^\top \mathbf{G}_t \mathbf{y}_t.$$

Similarly, we have that

$$\frac{d\|\mathbf{y}_t\|^2}{dt} = 2\mathbf{y}_t^\top \frac{d\mathbf{y}_t}{dt} = -2\mathbf{y}_t^\top \mathbf{g}_{\mathbf{y}_t} = -2\mathbf{y}_t^\top \mathbf{G}_t^\top \mathbf{x}_t.$$

Combining these two inequalities, we arrive at

$$\frac{d\|\mathbf{x}_t\|^2}{dt} - \frac{d\|\mathbf{y}_t\|^2}{dt} = 0.$$

The proof is thus completed. $\square$

## B.2 Extension to Stochastic Normalized Gradient Descent (SNGD)

Next, we extend Theorem 1 to SNGD, whose updates can be written as

$$\mathbf{x}_{t+1} = \mathbf{x}_t - \eta \frac{\mathbf{g}_{\mathbf{x}_t}}{\sqrt{\|\mathbf{g}_{\mathbf{x}_t}\|^2 + \|\mathbf{g}_{\mathbf{y}_t}\|^2}}, \qquad \mathbf{y}_{t+1} = \mathbf{y}_t - \eta \frac{\mathbf{g}_{\mathbf{y}_t}}{\sqrt{\|\mathbf{g}_{\mathbf{x}_t}\|^2 + \|\mathbf{g}_{\mathbf{y}_t}\|^2}}. \tag{15}$$

**Theorem 4.** *When applying SNGD (15) on NOP problem (1a), the limiting flow with $\eta \to 0$ guarantees that $\|\mathbf{x}_t\|^2 - \|\mathbf{y}_t\|^2 = \|\mathbf{x}_0\|^2 - \|\mathbf{y}_0\|^2$ for all $t > 0$. In other words, $\frac{d\mathcal{B}_t}{dt} = 0$ holds.*

*Proof.* For notational convenience, we let $\mathbf{G}_t := \nabla f_t(\mathbf{x}_t \mathbf{y}_t^\top)$. Then, we have that

$$\frac{d\|\mathbf{x}_t\|^2}{dt} = 2\mathbf{x}_t^\top \frac{d\mathbf{x}_t}{dt} = -2\frac{\mathbf{x}_t^\top \mathbf{g}_{\mathbf{x}_t}}{\sqrt{\|\mathbf{g}_{\mathbf{x}_t}\|^2 + \|\mathbf{g}_{\mathbf{y}_t}\|^2}} = -2\frac{\mathbf{x}_t^\top \mathbf{G}_t \mathbf{y}_t}{\sqrt{\|\mathbf{g}_{\mathbf{x}_t}\|^2 + \|\mathbf{g}_{\mathbf{y}_t}\|^2}}.$$

Similarly, we have that

$$\frac{d\|\mathbf{y}_t\|^2}{dt} = 2\mathbf{y}_t^\top \frac{d\mathbf{y}_t}{dt} = -2\frac{\mathbf{y}_t^\top \mathbf{g}_{\mathbf{y}_t}}{\sqrt{\|\mathbf{g}_{\mathbf{x}_t}\|^2 + \|\mathbf{g}_{\mathbf{y}_t}\|^2}} = -2\frac{\mathbf{y}_t^\top \mathbf{G}_t^\top \mathbf{x}_t}{\sqrt{\|\mathbf{g}_{\mathbf{x}_t}\|^2 + \|\mathbf{g}_{\mathbf{y}_t}\|^2}}.$$

Combining these two inequalities, we arrive at

$$\frac{d\|\mathbf{x}_t\|^2}{dt} - \frac{d\|\mathbf{y}_t\|^2}{dt} = 0.$$

The proof is thus completed. $\qquad\square$

## B.3 Proof of Theorem 2

*Proof.* Denote $\mathbf{G}_t = \nabla f_t(\mathbf{x}_t \mathbf{y}_t^\top)$ and $\tilde{\mathbf{G}}_t = \nabla f_t(\tilde{\mathbf{x}}_t \tilde{\mathbf{y}}_t^\top)$ for notational convenience. Following SAM updates in (4) and setting $\eta \to 0$, we have that

$$\frac{d\mathbf{x}_t}{dt} = -\tilde{\mathbf{G}}_t(\mathbf{y}_t + \rho u_t \mathbf{G}_t^\top \mathbf{x}_t), \quad \frac{d\mathbf{y}_t}{dt} = -\tilde{\mathbf{G}}_t^\top(\mathbf{x}_t + \rho u_t \mathbf{G}_t \mathbf{y}_t).$$

This gives that

$$\frac{1}{2}\frac{d\left(\|\mathbf{x}_t\|^2 - \|\mathbf{y}_t\|^2\right)}{dt} = \rho u_t \left[ \mathbf{y}_t^\top \tilde{\mathbf{G}}_t^\top \mathbf{G}_t \mathbf{y}_t - \mathbf{x}_t^\top \tilde{\mathbf{G}}_t \mathbf{G}_t^\top \mathbf{x}_t \right] \tag{16a}$$

$$= \rho u_t \left[ \|\mathbf{g}_{\mathbf{x}_t}\|^2 - \|\mathbf{g}_{\mathbf{y}_t}\|^2 \right] + \underbrace{\rho u_t \left[ \mathbf{y}_t^\top (\tilde{\mathbf{G}}_t - \mathbf{G}_t)^\top \mathbf{g}_{\mathbf{x}_t} - \mathbf{x}_t^\top (\tilde{\mathbf{G}}_t - \mathbf{G}_t) \mathbf{g}_{\mathbf{y}_t} \right]}_{:=\mathcal{A}_t}. \tag{16b}$$

The second term in (16b) is $\mathcal{A}_t$ in Theorem 2. Next, we give upper bound on $|\mathcal{A}_t|$. Using Assumption 1, we have that

$$\|\tilde{\mathbf{G}}_t - \mathbf{G}_t\| \le L\|\tilde{\mathbf{x}}_t \tilde{\mathbf{y}}_t^\top - \mathbf{x}_t \mathbf{y}_t^\top\|$$

$$= L\|\rho u_t(\mathbf{x}_t \mathbf{g}_{\mathbf{y}_t}^\top + \mathbf{g}_{\mathbf{x}_t} \mathbf{y}_t^\top) + \rho^2 u_t^2 \mathbf{g}_{\mathbf{x}_t} \mathbf{g}_{\mathbf{y}_t}^\top\|$$

$$\overset{(a)}{\le} L\rho \frac{\|\mathbf{x}_t \mathbf{g}_{\mathbf{y}_t}^\top + \mathbf{g}_{\mathbf{x}_t} \mathbf{y}_t^\top\|}{\sqrt{\|\mathbf{g}_{\mathbf{x}_t}\|^2 + \|\mathbf{g}_{\mathbf{y}_t}\|^2}} + L\rho^2 \frac{\|\mathbf{g}_{\mathbf{x}_t} \mathbf{g}_{\mathbf{y}_t}^\top\|}{\|\mathbf{g}_{\mathbf{x}_t}\|^2 + \|\mathbf{g}_{\mathbf{y}_t}\|^2}$$

$$\overset{(b)}{\le} L\rho(\|\mathbf{x}_t\| + \|\mathbf{y}_t\|) + \frac{L\rho^2}{2} = \mathcal{O}(L\rho)$$

where (a) uses the definition of $u_t$; (b) follows from $\|\mathbf{a}\mathbf{b}^\top\| = \|\mathbf{a}\|\|\mathbf{b}\|$ and the finite convergence assumption. To bound $\mathcal{A}_t$, we also have

$$\rho u_t |\mathbf{y}_t^\top (\tilde{\mathbf{G}}_t - \mathbf{G}_t)^\top \mathbf{g}_{\mathbf{x}_t}| = \rho \frac{|\mathbf{y}_t^\top (\tilde{\mathbf{G}}_t - \mathbf{G}_t)^\top \mathbf{g}_{\mathbf{x}_t}|}{\sqrt{\|\mathbf{g}_{\mathbf{x}_t}\|^2 + \|\mathbf{g}_{\mathbf{y}_t}\|^2}} \le \rho \frac{|\mathbf{y}_t^\top (\tilde{\mathbf{G}}_t - \mathbf{G}_t)^\top \mathbf{g}_{\mathbf{x}_t}|}{\|\mathbf{g}_{\mathbf{x}_t}\|}$$

$$\le \rho \|\tilde{\mathbf{G}}_t - \mathbf{G}_t\|\|\mathbf{y}_t\| = \mathcal{O}(L\rho^2) \tag{17}$$

where the last line also uses the finite convergence. We can bound $\rho u_t |\mathbf{x}_t^\top (\tilde{\mathbf{G}}_t - \mathbf{G}_t) \mathbf{g}_{\mathbf{y}_t}| = \mathcal{O}(\rho^2 L)$ in a similar manner. Combining (17) with (16b) gives the bound on $|\mathcal{A}_t| = \mathcal{O}(\rho^2 L)$. $\qquad\square$

## B.4 Proof of Corollary 1

Here, we prove the formal version of Corollary 1.

**Corollary 2.** *Suppose that $\|\mathbf{g}_{\mathbf{x}_t}\| > 0$ and $\|\mathbf{g}_{\mathbf{y}_t}\| > 0$ and $\rho \to 0$, then there exists $\bar{\mathcal{B}}_t$ such that the magnitude of $\mathcal{B}_t$ shrinks whenever $|\mathcal{B}_t| > \bar{\mathcal{B}}_t$.*

*Proof.* Without loss of generality, we suppose that $\mathcal{B}_t > 0$, i.e., $\|\mathbf{x}_t\| > \|\mathbf{y}_t\| > 0$. Let $\bar{\mathbf{x}}_t$ and $\bar{\mathbf{y}}_t$ be the scaled version of $\mathbf{x}_t$ and $\mathbf{y}_t$ such that $\|\bar{\mathbf{x}}_t\| = \|\bar{\mathbf{y}}_t\|$ and $\bar{\mathbf{x}}_t\bar{\mathbf{y}}_t^\top = \mathbf{x}_t\mathbf{y}_t^\top$ are satisfied. This suggests that $\mathbf{x}_t = \alpha_t\bar{\mathbf{x}}_t$ and $\mathbf{y}_t = \bar{\mathbf{y}}_t/\alpha_t$, where $\alpha_t = \sqrt{\|\mathbf{x}_t\|/\|\mathbf{y}_t\|}$. Next, we show that whenever $\mathcal{B}_t$ is large enough, we have that

$$\frac{\mathrm{d}\mathcal{B}_t}{\mathrm{d}t} = \rho\frac{\|\mathbf{g}_{\mathbf{x}_t}\|^2 - \|\mathbf{g}_{\mathbf{y}_t}\|^2}{\sqrt{\|\mathbf{g}_{\mathbf{x}_t}\|^2 + \|\mathbf{g}_{\mathbf{y}_t}\|^2}} + \mathcal{O}(\rho^2 L) < 0. \tag{18}$$

Since $\rho \to 0$, we only need to show that for some small $\epsilon = \mathcal{O}(\rho L) \geq 0$,

$$\frac{\|\mathbf{g}_{\mathbf{x}_t}\|^2 - \|\mathbf{g}_{\mathbf{y}_t}\|^2}{\sqrt{\|\mathbf{g}_{\mathbf{x}_t}\|^2 + \|\mathbf{g}_{\mathbf{y}_t}\|^2}} < -\epsilon. \tag{19}$$

By the definition of $\mathbf{g}_{\mathbf{x}_t}, \mathbf{g}_{\mathbf{y}_t}$ and $\bar{\mathbf{x}}_t, \bar{\mathbf{y}}_t$, we have that (19) can be rewritten as

$$\frac{\alpha_t^2\|\mathbf{G}_t^\top\bar{\mathbf{x}}_t\|^2 - \|\mathbf{G}_t\bar{\mathbf{y}}_t\|^2/\alpha_t^2}{\sqrt{\alpha_t^2\|\mathbf{G}_t^\top\bar{\mathbf{x}}_t\|^2 + \|\mathbf{G}_t\bar{\mathbf{y}}_t\|^2/\alpha_t^2}} > \epsilon. \tag{20}$$

Note that the function $h(z) := (az - b/z)/\sqrt{az + b/z}$ is monotonically increasing in $z$ when $a, b > 0$ and $z > 0$ as $h'(z) = (a^2z + 6ab/z + b^2/z^3)/(2(az + b/z)^{3/2}) > 0$. This implies that $h(z) > 0$ when $z > \sqrt{b/a}$, and thus the condition in (20) can be satisfied for $\epsilon = \mathcal{O}(\rho L) \to 0$ when $\alpha_t^2 > \bar{\alpha}^2$, where $\bar{\alpha}^2 := \|\mathbf{G}_t\bar{\mathbf{y}}_t\|/\|\mathbf{G}_t^\top\bar{\mathbf{x}}_t\|$. This condition on $\alpha_t$ is equivalent to

$$\begin{aligned}
\mathcal{B}_t &= \frac{1}{2}\big(\|\mathbf{x}_t\|^2 - \|\mathbf{y}_t\|^2\big) \\
&= \frac{1}{2}\big(\|\alpha_t\bar{\mathbf{x}}_t\|^2 - \|\bar{\mathbf{y}}_t/\alpha_t\|^2\big) \\
&> \frac{1}{2}\big(\|\bar{\alpha}\bar{\mathbf{x}}_t\|^2 - \|\bar{\mathbf{y}}_t/\bar{\alpha}\|^2\big).
\end{aligned}$$

Combining everything together, we have that $\frac{\mathrm{d}\mathcal{B}_t}{\mathrm{d}t} < 0$ if

$$\mathcal{B}_t > \bar{\mathcal{B}}_t := \frac{1}{2}\big(\|\bar{\alpha}\bar{\mathbf{x}}_t\|^2 - \|\bar{\mathbf{y}}_t/\bar{\alpha}\|^2\big). \tag{21}$$

The proof is thus completed. We also note that in the case of $\rho > 0$, the same condition as (21) can be derived by obtaining the inverse function of $h(z)$ evaluated at $\epsilon = \mathcal{O}(\rho L)$, and the corresponding $\bar{\alpha}_\rho$ and $\bar{\mathcal{B}}_t^\rho$ can be defined similarly. $\qquad\square$

## B.5 Extension to LoRA (layer-wise NOP problem)

Let $l \in \{1, 2, \ldots, D\}$ be the layer index. Denote $f_t$ as the loss function on minibatch $\mathcal{M}_t$. To simplify the notation, we also let $\mathbf{G}_{t,l} := \nabla_{\mathbf{x}_{t,l}\mathbf{y}_{t,l}^\top}f_t(\{\mathbf{x}_{t,l}, \mathbf{y}_{t,l}\}_l)$, $\tilde{\mathbf{G}}_{t,l} := \nabla_{\tilde{\mathbf{x}}_{t,l}\tilde{\mathbf{y}}_{t,l}^\top}f_t(\{\tilde{\mathbf{x}}_{t,l}, \tilde{\mathbf{y}}_{t,l}\}_l)$, and $u_t := 1/\sqrt{\sum_{l=1}^D\big(\|\mathbf{g}_{\mathbf{x}_{t,l}}\|^2 + \|\mathbf{g}_{\mathbf{y}_{t,l}}\|^2\big)}$. The update of SAM for layer $l$ can be written as

$$\tilde{\mathbf{x}}_{t,l} = \mathbf{x}_{t,l} + \rho u_t\mathbf{G}_{t,l}\mathbf{y}_{t,l}, \quad \tilde{\mathbf{y}}_{t,l} = \mathbf{y}_{t,l} + \rho u_t\mathbf{G}_{t,l}^\top\mathbf{x}_{t,l} \tag{22a}$$

$$\mathbf{g}_{\tilde{\mathbf{x}}_{t,l}} = \tilde{\mathbf{G}}_{t,l}\tilde{\mathbf{y}}_{t,l}, \quad \mathbf{g}_{\tilde{\mathbf{y}}_{t,l}} = \tilde{\mathbf{G}}_{t,l}^\top\tilde{\mathbf{x}}_{t,l} \tag{22b}$$

$$\mathbf{x}_{t+1,l} = \mathbf{x}_{t,l} - \eta\mathbf{g}_{\tilde{\mathbf{x}}_{t,l}}, \quad \mathbf{y}_{t+1,l} = \mathbf{y}_{t,l} - \eta\mathbf{g}_{\tilde{\mathbf{y}}_{t,l}}. \tag{22c}$$

**Refined assumption for LoRA.** Direct translating Assumption 1 to our multi-layer setting gives

$$\|\nabla f_t(\{\mathbf{x}_l\mathbf{y}_l^\top\}_l) - \nabla f_t(\{\mathbf{a}_l\mathbf{b}_l^\top\}_l)\|^2 \leq L^2\sum_{l=1}^D\|\mathbf{x}_l\mathbf{y}_l^\top - \mathbf{a}_l\mathbf{b}_l^\top\|^2. \tag{23}$$

However, the above assumption is loose, and our proof only needs block-wise smoothness, i.e.,

$$\|\nabla_l f_t(\mathbf{x}_l \mathbf{y}_l^\top) - \nabla_l f_t(\mathbf{a}_l \mathbf{b}_l^\top)\|^2 \leq \hat{L}^2 \|\mathbf{x}_l \mathbf{y}_l^\top - \mathbf{a}_l \mathbf{b}_l^\top\|^2, \forall l \tag{24}$$

where $\nabla_l$ refers to the gradient on $\mathbf{x}_l \mathbf{y}_l^\top$. It can be seen that $\sqrt{D}\hat{L} \geq L$, but one can assume that $\sqrt{D}\hat{L} \approx L$ for intuitive understandings.

**Theorem 5.** *Suppose that block smoothness assumption in* (24) *holds. Consider the limiting flow of SAM in* (22) *with* $\eta \to 0$ *and a sufficiently small* $\rho$. *Let* $\mathcal{B}_{t,l} := \frac{1}{2}\left(\|\mathbf{x}_{t,l}\|^2 - \|\mathbf{y}_{t,l}\|^2\right)$ *and* $\mathcal{B}_t = \sum_{l=1}^D \mathcal{B}_{t,l}$. *For some* $|\mathcal{A}_t| = \mathcal{O}(\rho^2 \hat{L})$, *SAM guarantees that*

$$\frac{d\mathcal{B}_t}{dt} = \rho \frac{\sum_{l=1}^D \|\mathbf{g}_{\mathbf{x}_{t,l}}\|^2 - \sum_{l=1}^D \|\mathbf{g}_{\mathbf{y}_{t,l}}\|^2}{\sqrt{\sum_{l=1}^D \|\mathbf{g}_{\mathbf{x}_{t,l}}\|^2 + \sum_{l=1}^D \|\mathbf{g}_{\mathbf{y}_{t,l}}\|^2}} + \mathcal{A}_t. \tag{25}$$

*Furthermore, for per layer balancedness it satisfies that for some* $|\mathcal{A}_{t,l}| = \mathcal{O}(\rho^2 \hat{L})$.

$$\frac{d\mathcal{B}_{t,l}}{dt} = \rho \frac{\|\mathbf{g}_{\mathbf{x}_{t,l}}\|^2 - \|\mathbf{g}_{\mathbf{y}_{t,l}}\|^2}{\sqrt{\sum_{l=1}^D \|\mathbf{g}_{\mathbf{x}_{t,l}}\|^2 + \sum_{l=1}^D \|\mathbf{g}_{\mathbf{y}_{t,l}}\|^2}} + \mathcal{A}_{t,i}. \tag{26}$$

**Understanding Theorem 5.** $\mathcal{A}_{t,i}$ and $\mathcal{A}_t$ are at the same order because of the possible unbalancedness among gradient norms for different layers. Comparing per layer balancedness $\mathcal{B}_{t,l}$ with Theorem 2, it can be roughly estimate that the regularization power is $\mathcal{O}(\sqrt{D})$ times smaller in $\mathcal{B}_{t,l}$. This estimation comes from $\hat{L} \approx L/\sqrt{D}$, and the first term is also $\mathcal{O}(\sqrt{D})$ smaller than the same term in Theorem 2. In other words, the regularization on balancedness can be reduced by $\mathcal{O}(\sqrt{D})$ times in LoRA in the worst case, and the worst case comes from gradient unbalancedness among layers.

*Proof.* Following (22) and setting $\eta \to 0$, we have that

$$\frac{d\mathbf{x}_{t,l}}{dt} = -\tilde{\mathbf{G}}_{t,l}(\mathbf{y}_{t,l} + \rho u_t \mathbf{G}_{t,l}^\top \mathbf{x}_{t,l}), \quad \frac{d\mathbf{y}_{t,l}}{dt} = -\tilde{\mathbf{G}}_{t,l}^\top(\mathbf{x}_{t,l} + \rho u_t \mathbf{G}_{t,l} \mathbf{y}_{t,l}).$$

This gives that

$$\frac{d\mathcal{B}_{t,l}}{dt} = \rho u_t \left[\mathbf{y}_{t,l}^\top \tilde{\mathbf{G}}_{t,l}^\top \mathbf{G}_{t,l} \mathbf{y}_{t,l} - \mathbf{x}_{t,l}^\top \tilde{\mathbf{G}}_{t,l} \mathbf{G}_{t,l}^\top \mathbf{x}_{t,l}\right] \tag{27a}$$

$$= \rho u_t \left[\|\mathbf{g}_{\mathbf{x}_{t,l}}\|^2 - \|\mathbf{g}_{\mathbf{y}_{t,l}}\|^2\right] + \underbrace{\rho u_t \left[\mathbf{y}_{t,l}^\top (\tilde{\mathbf{G}}_{t,l} - \mathbf{G}_{t,l})^\top \mathbf{g}_{\mathbf{x}_{t,l}} - \mathbf{x}_{t,l}^\top (\tilde{\mathbf{G}}_{t,l} - \mathbf{G}_{t,l}) \mathbf{g}_{\mathbf{y}_{t,l}}\right]}_{:=\mathcal{A}_{t,l}}. \tag{27b}$$

**Proof for** (25)**.** Let $\mathcal{A}_t := \sum_l \mathcal{A}_{t,l}$. To start with, we have that

$$\|\tilde{\mathbf{G}}_{t,l} - \mathbf{G}_{t,l}\| \leq \hat{L}\|\tilde{\mathbf{x}}_{t,l} \tilde{\mathbf{y}}_{t,l}^\top - \mathbf{x}_{t,l} \mathbf{y}_{t,l}^\top\|$$
$$= \hat{L}\|\rho u_t(\mathbf{x}_{t,l} \mathbf{g}_{\mathbf{y}_{t,l}}^\top + \mathbf{g}_{\mathbf{x}_{t,l}} \mathbf{y}_{t,l}^\top) + \rho^2 u_t^2 \mathbf{g}_{\mathbf{x}_{t,l}} \mathbf{g}_{\mathbf{y}_{t,l}}^\top\|$$

Next, based on finite convergence assumption, we have that

$$\rho u_t \sum_{l=1}^{D} \left| \mathbf{y}_{t,l}^\top (\tilde{\mathbf{G}}_{t,l} - \mathbf{G}_{t,l})^\top \mathbf{g}_{\mathbf{x}_{t,l}} \right| \tag{28}$$

$$\leq \sum_{l=1}^{D} \mathcal{O}\left( \rho u_t \|\tilde{\mathbf{G}}_{t,l} - \mathbf{G}_{t,l}\| \cdot \|\mathbf{g}_{\mathbf{x}_{t,l}}\| \right)$$

$$\overset{(a)}{\leq} \sum_{l=1}^{D} \mathcal{O}\left( \rho^2 u_t^2 \hat{L} \|\mathbf{x}_{t,l} \mathbf{g}_{\mathbf{y}_{t,l}}^\top + \mathbf{g}_{\mathbf{x}_{t,l}} \mathbf{y}_{t,l}^\top\| \cdot \|\mathbf{g}_{\mathbf{x}_{t,l}}\| \right)$$

$$\overset{(b)}{\leq} \sum_{l=1}^{D} \mathcal{O}\left( \rho^2 u_t^2 \hat{L} (\|\mathbf{g}_{\mathbf{y}_{t,l}}\| + \|\mathbf{g}_{\mathbf{x}_{t,l}}\|) \cdot \|\mathbf{g}_{\mathbf{x}_{t,l}}\| \right)$$

$$= \rho^2 \hat{L} \cdot \mathcal{O}\left( \frac{\sum_{l=1}^{D} \|\mathbf{g}_{\mathbf{x}_{t,l}}\|^2}{\sum_{l=1}^{D} (\|\mathbf{g}_{\mathbf{x}_{t,l}}\|^2 + \|\mathbf{g}_{\mathbf{y}_{t,l}}\|^2)} + \frac{\sum_{l=1}^{D} \|\mathbf{g}_{\mathbf{x}_{t,l}}\| \|\mathbf{g}_{\mathbf{y}_{t,l}}\|}{\sum_{l=1}^{D} (\|\mathbf{g}_{\mathbf{x}_{t,l}}\|^2 + \|\mathbf{g}_{\mathbf{y}_{t,l}}\|^2)} \right)$$

$$= \mathcal{O}(\rho^2 \hat{L})$$

where in (a) we use the fact that $\rho$ is chosen small; (b) uses finite convergence assumption and $\|\mathbf{a}\mathbf{b}^\top\| = \|\mathbf{a}\| \|\mathbf{b}\|$. Using similar arguments, we can bound $\mathcal{A}_t = \mathcal{O}(\rho^2 \hat{L})$.

**Proof for** (26). Next, we give upper bound on $|\mathcal{A}_{t,l}|$. Using similar argument as (28), we have that

$$\rho u_t \left| \mathbf{y}_{t,l}^\top (\tilde{\mathbf{G}}_{t,l} - \mathbf{G}_{t,l})^\top \mathbf{g}_{\mathbf{x}_{t,l}} \right| \tag{29}$$

$$\leq \mathcal{O}\left( \rho^2 u_t^2 \hat{L} (\|\mathbf{g}_{\mathbf{y}_{t,l}}\| + \|\mathbf{g}_{\mathbf{x}_{t,l}}\|) \cdot \|\mathbf{g}_{\mathbf{x}_{t,l}}\| \right)$$

$$= \rho^2 \hat{L} \cdot \mathcal{O}\left( \frac{\|\mathbf{g}_{\mathbf{x}_{t,l}}\|^2}{\sum_{l=1}^{D} (\|\mathbf{g}_{\mathbf{x}_{t,l}}\|^2 + \|\mathbf{g}_{\mathbf{y}_{t,l}}\|^2)} + \frac{\|\mathbf{g}_{\mathbf{x}_{t,l}}\| \|\mathbf{g}_{\mathbf{y}_{t,l}}\|}{\sum_{l=1}^{D} (\|\mathbf{g}_{\mathbf{x}_{t,l}}\|^2 + \|\mathbf{g}_{\mathbf{y}_{t,l}}\|^2)} \right). \tag{30}$$

Using (29), we have that

$$|\mathcal{A}_{t,l}| \leq \rho^2 \hat{L} \cdot \mathcal{O}\left( \frac{\|\mathbf{g}_{\mathbf{x}_{t,l}}\|^2 + \|\mathbf{g}_{\mathbf{y}_{t,l}}\|^2}{\sum_{l=1}^{D} (\|\mathbf{g}_{\mathbf{x}_{t,l}}\|^2 + \|\mathbf{g}_{\mathbf{y}_{t,l}}\|^2)} + \frac{\|\mathbf{g}_{\mathbf{x}_{t,l}}\| \|\mathbf{g}_{\mathbf{y}_{t,l}}\|}{\sum_{l=1}^{D} (\|\mathbf{g}_{\mathbf{x}_{t,l}}\|^2 + \|\mathbf{g}_{\mathbf{y}_{t,l}}\|^2)} \right)$$

$$= \mathcal{O}(\rho^2 \hat{L}).$$

The proof is is thus completed. $\square$

## C  Missing Proofs for OP

### C.1  Unbalancedness of SGD in OP

**Theorem 6.** *Applied SGD or SNGD on problem* (1b)*, both of them ensure that* $\|\mathbf{x}_t\|^2 - \|\mathbf{y}_t\|^2 = \|\mathbf{x}_0\|^2 - \|\mathbf{y}_0\|^2$ *for all* $t > 0$*. In other words,* $\mathcal{B}_t$ *keeps unchanged.*

*Proof.* We consider SGD and NSGD separately.

**SGD.** It is straightforward to see that

$$\frac{d\|\mathbf{x}_t\|^2}{dt} = -2f_t'(\mathbf{x}_t^\top \mathbf{y}_t) \mathbf{x}_t^\top \mathbf{y}_t = \frac{d\|\mathbf{y}_t\|^2}{dt}.$$

This completes the proof of SGD.

**NSGD.** The gradient update of NSGD is

$$\frac{d\mathbf{x}_t}{dt} = -\frac{\mathbf{g}_{\mathbf{x}_t}}{\sqrt{\|\mathbf{g}_{\mathbf{x}_t}\|^2 + \|\mathbf{g}_{\mathbf{y}_t}\|^2}}, \quad \frac{d\mathbf{y}_t}{dt} = -\frac{\mathbf{g}_{\mathbf{y}_t}}{\sqrt{\|\mathbf{g}_{\mathbf{x}_t}\|^2 + \|\mathbf{g}_{\mathbf{y}_t}\|^2}}. \tag{31}$$

Then we have that for NSGD,

$$\frac{d\|\mathbf{x}_t\|^2}{dt} = -2f_t'(\mathbf{x}_t^\top \mathbf{y}_t) \frac{\mathbf{x}_t^\top \mathbf{y}_t}{\sqrt{\|\mathbf{g}_{\mathbf{x}_t}\|^2 + \|\mathbf{g}_{\mathbf{y}_t}\|^2}} = \frac{d\|\mathbf{y}_t\|^2}{dt}.$$

This gives the result for SNGD. $\square$

## C.2  Proof of Theorem 3

To prove this theorem, we first focus on the dynamic of SAM.

**Lemma 2.** *Suppose that Assumption 1 holds. Consider the limiting flow of SAM in (7) with $\eta \to 0$. Let $\mathcal{B}_t := \frac{1}{2}\left(\|\mathbf{x}_t\|^2 - \|\mathbf{y}_t\|^2\right)$ and $\rho$ be small. Then, for some $|\mathcal{A}_t| = \mathcal{O}(\rho^2 L|\mathcal{B}_t|)$, SAM guarantees*

$$\frac{d\mathcal{B}_t}{dt} = -2\rho \frac{|f_t'(\mathbf{x}_t^\top \mathbf{y}_t)|}{\sqrt{\|\mathbf{x}_t\|^2 + \|\mathbf{y}_t\|^2}} \mathcal{B}_t + \mathcal{A}_t. \tag{32}$$

*Proof.* For notational convenience, we write $f_t' := f_t'(\mathbf{x}_t^\top \mathbf{y}_t)$ and $\tilde{f}_t' := f_t'(\tilde{\mathbf{x}}_t^\top \tilde{\mathbf{y}}_t)$. Using similar arguments as Theorem 2, we have that

$$\frac{1}{2}\frac{d}{dt}\left(\|\mathbf{x}_t\|^2 - \|\mathbf{y}_t\|^2\right) = -\rho u_t \tilde{f}_t' \cdot \left(\|\mathbf{x}_t\|^2 - \|\mathbf{y}_t\|^2\right) \tag{33}$$

$$= -\rho \frac{\operatorname{sgn}(f_t')\tilde{f}_t'}{\sqrt{\|\mathbf{x}_t\|^2 + \|\mathbf{y}_t\|^2}} \cdot \left(\|\mathbf{x}_t\|^2 - \|\mathbf{y}_t\|^2\right)$$

$$= -\rho \frac{|f_t'|}{\sqrt{\|\mathbf{x}_t\|^2 + \|\mathbf{y}_t\|^2}} \cdot \left(\|\mathbf{x}_t\|^2 - \|\mathbf{y}_t\|^2\right)$$

$$+ \underbrace{\rho \frac{\operatorname{sgn}(f_t')(f_t' - \tilde{f}_t')}{\sqrt{\|\mathbf{x}_t\|^2 + \|\mathbf{y}_t\|^2}} \cdot \left(\|\mathbf{x}_t\|^2 - \|\mathbf{y}_t\|^2\right)}_{:=\mathcal{A}_t}.$$

Next we bound $|\mathcal{A}_t|$. To start with, we have that

$$\left|\tilde{\mathbf{x}}_t^\top \tilde{\mathbf{y}}_t - \mathbf{x}_t^\top \mathbf{y}_t\right| = \left|\rho^2 u_t^2 \mathbf{x}_t^\top \mathbf{y}_t + \rho u_t \|\mathbf{x}_t\|^2 + \rho u_t \|\mathbf{y}_t\|^2\right| \tag{34}$$

$$\leq \rho^2 \frac{|\mathbf{x}_t^\top \mathbf{y}_t|}{\|\mathbf{x}_t\|^2 + \|\mathbf{y}_t\|^2} + \rho\sqrt{\|\mathbf{x}_t\|^2 + \|\mathbf{y}_t\|^2}$$

$$\leq \frac{\rho^2}{2} + \rho\sqrt{\|\mathbf{x}_t\|^2 + \|\mathbf{y}_t\|^2}.$$

Using Assumption 1 and (34), we arrive at

$$|f_t' - \tilde{f}_t'| \leq L\left|\tilde{\mathbf{x}}_t^\top \tilde{\mathbf{y}}_t - \mathbf{x}_t^\top \mathbf{y}_t\right| = \mathcal{O}(\rho L\sqrt{\|\mathbf{x}_t\|^2 + \|\mathbf{y}_t\|^2}). \tag{35}$$

Hence, we arrive at

$$|\mathcal{A}_t| \leq \rho|f_t' - \tilde{f}_t'|\left|\frac{\|\mathbf{x}_t\|^2 - \|\mathbf{y}_t\|^2}{\sqrt{\|\mathbf{x}_t\|^2 + \|\mathbf{y}_t\|^2}}\right| = \mathcal{O}(\rho^2 L|\mathcal{B}_t|).$$

The proof is thus completed. $\qquad\square$

Next, the proof of Theorem 3 is provided.

*Proof.* Lemma 2 has already indicated the concentration of $\mathcal{B}_t$ towards 0, if the magnitude of the first term is larger than $|\mathcal{A}_t|$. To see this, notice that we can lower bound $2|\mathcal{B}_t|/\sqrt{\|\mathbf{x}_t\|^2 + \|\mathbf{y}_t\|^2}$ by

$$\left|\frac{\|\mathbf{x}_t\|^2 - \|\mathbf{y}_t\|^2}{\sqrt{\|\mathbf{x}_t\|^2 + \|\mathbf{y}_t\|^2}}\right| = \left|\frac{(\|\mathbf{x}_t\| + \|\mathbf{y}_t\|)(\|\mathbf{x}_t\| - \|\mathbf{y}_t\|)}{\sqrt{\|\mathbf{x}_t\|^2 + \|\mathbf{y}_t\|^2}}\right| \geq \left|\|\mathbf{x}_t\| - \|\mathbf{y}_t\|\right| = \mathcal{C}_t. \tag{36}$$

Hence, long as $\rho|f_t'(\mathbf{x}_t^\top \mathbf{y}_t)| \cdot \mathcal{C}_t > \mathcal{O}(\rho^2 L|\mathcal{B}_t|)$, we have the first term dominating the dynamic of SAM, leading to contraction of $\mathcal{B}_t$. This completes the proof to the first part.

Next we prove the second part, which is the lower- and upper- bound on $\mathcal{B}_t$. The lower bound can be seen from (36). For the upper bound, we have

$$\left|\frac{\|\mathbf{x}_t\|^2 - \|\mathbf{y}_t\|^2}{\sqrt{\|\mathbf{x}_t\|^2 + \|\mathbf{y}_t\|^2}}\right| \leq \left|\frac{\|\mathbf{x}_t\|^2 - \|\mathbf{y}_t\|^2}{\sqrt{\|\mathbf{x}_t\|^2 - \|\mathbf{y}_t\|^2}}\right| = \sqrt{2|\mathcal{B}_t|}. \tag{37}$$

Plugging (37) into (33) finishes the proof. $\qquad\square$

## C.3 $m$-sharpness for OP

$m$-sharpness is a variant of SAM that is empirically observed to improve generalization, and it is especially useful for distributed training on multiple GPUs (Foret et al., 2021). However, the reason behind the improved performance is not fully understood. (Andriushchenko and Flammarion, 2022) show that $m$-sharpness is more sparse-promoting for diagonal linear neural networks minimized via a quadratic loss. However, diagonal linear networks are not scale-invariant.

For consistent notation with (7), we use $f_t(\cdot)$ to denote the loss function on minibatch $\mathcal{M}_t$. In $m$-sharpness, the minibatch $\mathcal{M}_t$ is divided into $m$ disjoint subsets. Without loss of generality, we also assume that the minibatch is evenly divided. We denote the loss function on each subset as $f_{t,i}, i \in \{1, 2, \ldots, m\}$. Note that we have $\frac{1}{m} \sum_{i=1}^m f_{t,i} = f_t$. With these definitions, the update of $m$-sharpness can be written as

$$\tilde{\mathbf{x}}_{t,i} = \mathbf{x}_t + \rho u_{t,i} \mathbf{y}_t, \quad \tilde{\mathbf{y}}_{t,i} = \mathbf{y}_t + \rho u_{t,i} \mathbf{x}_t \tag{38a}$$

$$\mathbf{g}^i_{\tilde{\mathbf{x}}_{t,i}} = f'_{t,i}(\tilde{\mathbf{x}}_{t,i}^\top \tilde{\mathbf{y}}_{t,i}) \tilde{\mathbf{y}}_{t,i}, \quad \mathbf{g}^i_{\tilde{\mathbf{y}}_{t,i}} = f'_{t,i}(\tilde{\mathbf{x}}_{t,i}^\top \tilde{\mathbf{y}}_{t,i}) \tilde{\mathbf{x}}_{t,i} \tag{38b}$$

$$\mathbf{x}_{t+1} = \mathbf{x}_t - \eta \frac{1}{m} \sum_{i=1}^m \mathbf{g}^i_{\tilde{\mathbf{x}}_{t,i}}, \quad \mathbf{y}_{t+1} = \mathbf{y}_t - \eta \frac{1}{m} \sum_{i=1}^m \mathbf{g}^i_{\tilde{\mathbf{y}}_{t,i}}. \tag{38c}$$

where $u_{t,i} := \operatorname{sgn}(f'_{t,i}(\mathbf{x}_t^\top \mathbf{y}_t))/\sqrt{\|\mathbf{x}_t\|^2 + \|\mathbf{y}_t\|^2}$. Comparing with the SAM update for OP in (7), the difference is that perturbed gradient is calculated on each $f_{t,i}$. Next, we analyze the dynamic of SAM with $m$-sharpness.

**Lemma 3.** *Suppose that Assumption 1 holds. Consider the limiting flow of SAM in (38) with $\eta \to 0$. Let $\mathcal{B}_t := \frac{1}{2}(\|\mathbf{x}_t\|^2 - \|\mathbf{y}_t\|^2)$ and $\rho$ be small. Then, for some $|\mathcal{A}_t| = \mathcal{O}(\rho^2 L)$, SAM guarantees that*

$$\frac{d\mathcal{B}_t}{dt} = -2 \frac{\rho}{m} \frac{\sum_{i=1}^m |f'_{t,i}(\mathbf{x}_t^\top \mathbf{y}_t)|}{\sqrt{\|\mathbf{x}_t\|^2 + \|\mathbf{y}_t\|^2}} \mathcal{B}_t + \mathcal{A}_t. \tag{39}$$

*Proof.* For notational convenience, we write $f'_{t,i} := f'_{t,i}(\mathbf{x}_t^\top \mathbf{y}_t)$ and $\tilde{f}'_{t,i} := f'_{t,i}(\tilde{\mathbf{x}}_{t,i}^\top \tilde{\mathbf{y}}_{t,i})$. Then, we have that

$$\begin{aligned} \frac{1}{2} \frac{\mathrm{d}}{\mathrm{d}t}\left(\|\mathbf{x}_t\|^2 - \|\mathbf{y}_t\|^2\right) &= -\frac{\rho}{m} \sum_{i=1}^m u_{t,i} \tilde{f}'_{t,i} \cdot \left(\|\mathbf{x}_t\|^2 - \|\mathbf{y}_t\|^2\right) \\ &= -\frac{\rho}{m} \sum_{i=1}^m \frac{\operatorname{sgn}(f'_{t,i}) \tilde{f}'_{t,i}}{\sqrt{\|\mathbf{x}_t\|^2 + \|\mathbf{y}_t\|^2}} \cdot \left(\|\mathbf{x}_t\|^2 - \|\mathbf{y}_t\|^2\right) \\ &= -\frac{\rho}{m} \frac{\sum_{i=1}^m |f'_{t,i}|}{\sqrt{\|\mathbf{x}_t\|^2 + \|\mathbf{y}_t\|^2}} \cdot \left(\|\mathbf{x}_t\|^2 - \|\mathbf{y}_t\|^2\right) \\ &\quad + \frac{\rho}{m} \sum_{i=1}^m \underbrace{\frac{\operatorname{sgn}(f'_{t,i})(f'_{t,i} - \tilde{f}'_{t,i})}{\sqrt{\|\mathbf{x}_t\|^2 + \|\mathbf{y}_t\|^2}} \cdot \left(\|\mathbf{x}_t\|^2 - \|\mathbf{y}_t\|^2\right)}_{:=\mathcal{A}_{t,i}}. \end{aligned} \tag{40}$$

Next, using (34) and Assumption 1, we have

$$|f'_{t,i} - \tilde{f}'_{t,i}| \leq L|\tilde{\mathbf{x}}_{t,i}^\top \tilde{\mathbf{y}}_{t,i} - \mathbf{x}_t^\top \mathbf{y}_t| = \mathcal{O}(\rho L \sqrt{\|\mathbf{x}_t\|^2 + \|\mathbf{y}_t\|^2}).$$

Hence, we can bound $|\mathcal{A}_{t,i}|$ as

$$|\mathcal{A}_{t,i}| \leq |f'_{t,i} - \tilde{f}'_{t,i}| \left| \frac{\|\mathbf{x}_t\|^2 - \|\mathbf{y}_t\|^2}{\sqrt{\|\mathbf{x}_t\|^2 + \|\mathbf{y}_t\|^2}} \right| = \mathcal{O}(\rho L |\mathcal{B}_t|).$$

The proof is thus completed by plugging $|\mathcal{A}_{t,i}|$ into (40). $\qquad\square$

## C.4 Extension to Layer-wise OP

We start with the notation. Let $l \in \{1, 2, \ldots, D\}$ be the layer index. Denote $f_t$ as the loss on minibatch $\mathcal{M}_t$. Let $f'_{t,l} := \nabla_l f_t(\{\mathbf{x}_{t,l}^\top \mathbf{y}_{t,l}\}_l)$, i.e., the $l$-th entry of gradient (w.r.t. the variable $\mathbf{x}_{t,l}^\top \mathbf{y}_{t,l}$), $\tilde{f}'_{t,l} := \nabla_l f_t(\{\tilde{\mathbf{x}}_{t,l}^\top \tilde{\mathbf{y}}_{t,l}\}_l)$, and $u_t := 1/\sqrt{\sum_{l=1}^D |f'_{t,l}|^2 [\|\mathbf{x}_{t,l}\|^2 + \|\mathbf{y}_{t,l}\|^2]}$. The update of SAM for layer $l$ can be written as

$$\tilde{\mathbf{x}}_{t,l} = \mathbf{x}_{t,l} + \rho u_t f'_{t,l} \mathbf{y}_{t,l}, \quad \tilde{\mathbf{y}}_{t,l} = \mathbf{y}_{t,l} + \rho u_t f'_{t,l} \mathbf{x}_{t,l}, \tag{41a}$$

$$\mathbf{g}_{\tilde{\mathbf{x}}_{t,l}} = \tilde{f}'_{t,l} \tilde{\mathbf{y}}_{t,l}, \quad \mathbf{g}_{\tilde{\mathbf{y}}_{t,l}} = \tilde{f}'_{t,l} \tilde{\mathbf{x}}_{t,l} \tag{41b}$$

$$\mathbf{x}_{t+1,l} = \mathbf{x}_{t,l} - \eta \mathbf{g}_{\tilde{\mathbf{x}}_{t,l}}, \quad \mathbf{y}_{t+1,l} = \mathbf{y}_{t,l} - \eta \mathbf{g}_{\tilde{\mathbf{y}}_{t,l}}. \tag{41c}$$

**Refined assumption for LoRA.** Our proof only needs block-wise smoothness, i.e.,

$$|\nabla_l f_t(\mathbf{x}_l^\top \mathbf{y}_l) - \nabla_l f_t(\mathbf{a}_l^\top \mathbf{b}_l)|^2 \leq \hat{L}^2 |\mathbf{x}_l^\top \mathbf{y}_l - \mathbf{a}_l^\top \mathbf{b}_l|^2, \ \forall l, \tag{42}$$

where $\nabla_l$ refers to the gradient on $\mathbf{x}_l^\top \mathbf{y}_l$. It can be seen that $\sqrt{D}\hat{L} \geq L$, but one can assume that $\sqrt{D}\hat{L} \approx L$ for more clear intuition.

**Theorem 7.** *Suppose that block smoothness assumption in* (42) *holds. Consider the limiting flow of SAM in* (41) *with* $\eta \to 0$ *and a sufficiently small* $\rho$. *Let* $\mathcal{B}_{t,l} := \frac{1}{2}(\|\mathbf{x}_{t,l}\|^2 - \|\mathbf{y}_{t,l}\|^2)$ *and* $\mathcal{B}_t^{\max} = \max_l |\mathcal{B}_{t,l}|$. *For some* $|\mathcal{A}_t| = \mathcal{O}(\rho^2 \hat{L} \mathcal{B}_t^{\max})$, *SAM guarantees that*

$$\frac{d\mathcal{B}_t}{dt} = -\rho \frac{\sum_{l=1}^D |f'_{t,l}|^2 (\|\mathbf{x}_{t,l}\|^2 - \|\mathbf{y}_{t,l}\|^2)}{\sqrt{\sum_{l=1}^D |f'_{t,l}|^2 [\|\mathbf{x}_{t,l}\|^2 + \|\mathbf{y}_{t,l}\|^2]}} + \mathcal{A}_t. \tag{43}$$

*Furthermore, for some* $|\mathcal{A}_{t,l}| = \mathcal{O}(\rho^2 \hat{L} |\mathcal{B}_{t,l}|)$, *per layer balancedness satisfies that*

$$\frac{d\mathcal{B}_{t,l}}{dt} = -\rho \frac{|f'_{t,l}|^2 (\|\mathbf{x}_{t,l}\|^2 - \|\mathbf{y}_{t,l}\|^2)}{\sqrt{\sum_{l=1}^D |f'_{t,l}|^2 [\|\mathbf{x}_{t,l}\|^2 + \|\mathbf{y}_{t,l}\|^2]}} + \mathcal{A}_{t,i}. \tag{44}$$

*Proof.* Using a similar derivation as before, we have that

$$\frac{1}{2} \frac{d}{dt}\left(\|\mathbf{x}_{t,l}\|^2 - \|\mathbf{y}_{t,l}\|^2\right) = -\rho u_t |f'_{t,l}|^2 \cdot \left(\|\mathbf{x}_{t,l}\|^2 - \|\mathbf{y}_{t,l}\|^2\right)$$

$$+ \underbrace{\rho u_t f'_{t,l}(f'_{t,l} - \tilde{f}'_{t,l}) \cdot \left(\|\mathbf{x}_{t,l}\|^2 - \|\mathbf{y}_{t,l}\|^2\right)}_{:=\mathcal{A}_{t,l}}$$

Next, based on (42), we have that

$$|f'_{t,l} - \tilde{f}'_{t,l}| \leq \hat{L} |\tilde{\mathbf{x}}_{t,l}^\top \tilde{\mathbf{y}}_{t,l} - \mathbf{x}_{t,l}^\top \mathbf{y}_{t,l}| \leq \rho \hat{L} u_t |f'_{t,l}|(\|\mathbf{x}_{t,l}\|^2 + \|\mathbf{y}_{t,l}\|^2) + \rho^2 \hat{L} u_t^2 |f'_{t,l}|^2 |\mathbf{x}_{t,l}^\top \mathbf{y}_{t,l}|.$$

Combining these two equations, and applying similar argument as Theorem 5, it is not difficult to arrive at $|\mathcal{A}_{t,i}| = \mathcal{O}(\rho^2 \hat{L} |\mathcal{B}_{t,l}|)$ and $|\mathcal{A}_t| = \mathcal{O}(\rho^2 \hat{L} \mathcal{B}_t^{\max})$. $\qquad \square$

## C.5 Proof of Lemma 1

*Proof.* Within $\mathcal{W}^*$, the Hessian on $(\mathbf{x}, \mathbf{y})$ can be calculated as $f''(\mathbf{x}^\top \mathbf{y})[\mathbf{y}^\top, \mathbf{x}^\top]^\top [\mathbf{y}^\top, \mathbf{x}^\top]$. The largest eigenvalue is $f''(w)(\|\mathbf{x}\|^2 + \|\mathbf{y}\|^2)$. By the AM-GM inequality, it can be seen that the largest eigenvalue is minimized when $\|\mathbf{x}\| = \|\mathbf{y}\|$, whose balancedness is 0. $\qquad \square$

# D Missing Experimental Details

We mainly focus on finetuning LMs with LoRA. This setting naturally includes distributional shift – the finetuning dataset does not usually have the same distribution as the pretraining dataset as validated through zero-shot performance. All experiments are performed on a server with AMD EPYC 7742 CPUs and NVIDIA GeForce RTX 3090 GPUs each with 24GiB memory. All numerical results from Section 6 report test performance (e.g., accuracy, F1 scores, or BLEU scores) and the standard deviation across multiple runs.

## D.1 Details on Datasets

Our evaluations are carried out on commonly-used datasets in the literature.

**GLUE benchmark.** GLUE is designed to provide a general-purpose evaluation of language understanding (Wang et al., 2019b). Those adopted in our work include MNLI (inference, (Williams et al., 2018)), SST-2 (sentiment analysis, (Socher et al., 2013)), MRPC (paraphrase detection, (Dolan and Brockett, 2005)), CoLA (linguistic acceptability (Warstadt et al., 2019)), QNLI (inference (Rajpurkar et al., 2018)), QQP[3] (question-answering), RTE[4] (inference), and STS-B (textual similarity (Cer et al., 2017)). These datasets are released under different permissive licenses.

**SuperGLUE benchmark.** SuperGLUE (Wang et al., 2019a) is another commonly adopted benchmark for language understanding and is more challenging compared with GLUE. The considered datasets include CB (inference, (De Marneffe et al., 2019)), ReCoRD (multiple-choice question answering (Zhang et al., 2018)), COPA (question answering (Roemmele et al., 2011)). These datasets are released under different permissive licenses.

**WebNLG Challenge.** This dataset is commonly used for data-to-text evaluation (Gardent et al., 2017). It has 22K examples in total with 14 distinct categories. Among them, 9 are seen during training, and the unseen training data are used to test the generalization performance. The dataset is released under license CC BY-NC-SA 4.0.

**Additional datasets.** We also use SQuAD (question answering (Rajpurkar et al., 2016)) in our experiments, which is released under license CC BY-SA 4.0. Other datasets include TREC (topic classification (Voorhees and Tice, 2000)) and SNLI (inference (Bowman et al., 2015)). Both of them are licensed under CC BY-SA 4.0.

## D.2 Details on Language Models

We summarize the adopted language models in our evaluation. All model checkpoints are obtained from HuggingFace.

**RoBERTa-large.** This is a 355M parameter model. The model checkpoint[5] is released under the MIT license.

**OPT-1.3B.** The model checkpoint[6] is released under a non-commercial license. [7]

**GPT2-medium.** This is a 345M parameter model. Its checkpoint[8] is under MIT License.

## D.3 Few-shot Learning with RoBERTa and OPT

**Experiments on RoBERTa-large.** We follow the $k$-shot learning setup in (Malladi et al., 2023) and focus on classification tasks. The training set contains $k = 512$ samples per class while the test set has 1000 samples. We also employ prompts for finetuning; where the adopted prompts are the same as those in (Malladi et al., 2023, Table 13). AdamW is adopted as the base optimizer, and hyperparameters are tuned from those in Table 6. Our experiments are averaged over 3 random trials. The estimated runtime is about 5 minutes per dataset.

The per-iteration runtime on the SST-5 dataset of BAR, SAM, and the baseline optimizer are compared in Table 7. It can be seen that SAM is much more slower than the baseline approach, and BAR reduces 74% additional runtime of SAM, while achieving comparable accuracy. We believe that this runtime saving can be even larger with additional engineering efforts such as kernel fusion, which we leave for future work. This validates the computational efficiency of BAR.

---

[3] https://quoradata.quora.com/First-Quora-Dataset-Release-Question-Pairs
[4] https://paperswithcode.com/dataset/rte
[5] https://huggingface.co/FacebookAI/roberta-large
[6] https://huggingface.co/facebook/opt-1.3b
[7] https://github.com/facebookresearch/metaseq/blob/main/projects/OPT/MODEL_LICENSE.md
[8] https://s3.amazonaws.com/models.huggingface.co/bert/gpt2-medium-pytorch_model.bin

Table 6: Hyperparameters used for few-shot learning with RoBERTa-large.

| Hyper-parameters | Values |
|---|---|
| LoRA $r$ (rank) | 8 |
| LoRA $\alpha$ | 16 |
| # iterations | 1000 |
| batchsize | 16 |
| learning rate | $1\times10^{-4}, 3\times10^{-4}, 5\times10^{-4}$ |
| $\rho$ for SAM | 0.05, 0.1, 0.2 |
| $\mu_0$ for BAR | 0.5, 1.0, 2.0 |
| scheduler for BAR | linear, cosine |

Table 7: Per-iteration runtime for finetuning RoBERTa-large on SST5.

| SST5 | baseline | SAM | BAR |
|---|---|---|---|
| time (s) | 0.105 | 0.265 | 0.146 |

**Experiments on OPT.** For OPT-1.3B, we consider tasks from the SuperGLUE benchmark covering classification and multiple-choice. We also consider generation tasks on SQuAD. Following (Malladi et al., 2023), we randomly sample 1000 data for training and the other 1000 for testing. AdamW is adopted as base optimizer. The hyperparameters adopted are searched over values in Table 8. Estimated runtime is less than or around 10 minutes, depending on the dataset.

If we directly apply FP16 training with SAM, *underflow* can happen if one does not take care of the gradient scaling on the two gradients calculated per iteration. This means that SAM is not flexible enough to be integrated with the codebase for large scale training, as FP16 is the default choice for finetuning LMs. We employ FP32 to bypass the issue with SAM. Consequently, the training speed is significantly slowed down; see a summary in Table 9. It further demonstrates the effectiveness of BAR for large scale-training.

Overall, the results for few-shot learning indicate that given limited data, BAR can effectively improve generalization using significantly reduced computational resources relative to SAM.

Table 8: Hyperparameters used for few-shot learning with OPT-1.3B.

| Hyper-parameters | Values |
|---|---|
| LoRA $r$ (rank) | 8 |
| LoRA $\alpha$ | 16 |
| # iterations | 1000 |
| batchsize | 2, 4, 8 |
| learning rate | $1\times10^{-5}, 1\times10^{-4}, 5\times10^{-4}$ |
| $\rho$ for SAM | 0.05, 0.1, 0.2 |
| $\mu_0$ for BAR | 0.2, 0.5, 1.0, 2.0 |
| scheduler for BAR | linear, cosine |

### D.4 Finetuning with RoBERTa-large

Our implementation is inspired from (Hu et al., 2022)[9], which is under MIT License. The hyper-parameters are chosen the same as provided in its GitHub Repo. AdamW is adopted as the base optimizer. However, we employ single GPU rather than multiple ones and use gradient accumulation rather than parallelism due to memory constraint. We also note that there could be failure cases for LoRA using certain seed, e.g., SST-2 with seed 1 and MNLI with seed 2. These cases are ignored when comparing. We consider the GLUE benchmark and report the mismatched accuracy for MNLI, Matthew's correlation for CoLA, Pearson correlation for STS-B, and accuracy for other datasets. Larger values indicate better results for all datasets. For LoRA, we employ $r = 8$ and $\alpha = 16$.

---

[9]https://github.com/microsoft/LoRA/tree/main

Table 9: Per-iteration runtime for finetuning OPT-1.3B on RTE.

| RTE | baseline | SAM | BAR |
|---|---|---|---|
| precision | FP16 | FP32 | FP16 |
| time (s) | 0.1671 | 0.708 | 0.1731 |

Table 10: Experiments on finetuning RoBERTa (355M). Results marked with † are taken from (Hu et al., 2022), and those with * refer to Adapter[P] in (Hu et al., 2022).

| RoBERTa | # para | SST2 | STS-B | RTE | QQP | QNLI | MRPC | MNLI | CoLA | avg |
|---|---|---|---|---|---|---|---|---|---|---|
| FT[†] | 355M | 96.4 | 92.4 | 86.6 | 92.2 | 94.7 | 90.9 | 90.2 | 68.0 | 88.9 |
| Adapter* | 0.8M | **96.6** | 91.9 | 80.1 | **91.7** | **94.8** | 89.7 | - | **67.8** | - |
| LoRA | 0.8M | 95.8 | 92.4 | 88.2 | 91.4 | 94.7 | 89.6 | 90.6 | 64.8 | 88.4 |
| **LoRA-oBAR** | 0.8M | 96.0 | **92.6** | 88.7 | 91.6 | **94.8** | **90.3** | 90.6 | 65.1 | 88.7 |
| **LoRA-nBAR** | 0.8M | 96.0 | **92.6** | **89.2** | 91.6 | 94.7 | **90.3** | **90.8** | 65.6 | **88.9** |

Experiments are conducted over three random trials for all datasets, with the exception of QQP, for which only two trials are performed due to its large size. The results of final test performance can be found in Table 10. Estimated runtime varies for different datasets from 2 to 15 hours, except for QQP which takes 3 days on our device.

For the hyperparameters of oBAR and nBAR, $\mu_0$ is typically chosen from $\{0.2, 0.5, 1.0\}$; however, for QQP, a value of $0.05$ is used. The scheduler is chosen from linear and constant. We also observe that for datasets such as COLA and RTE, setting weight decay as $0$ works best for BAR.

### D.5 GPT2 medium on WebNLG Challenge

AdamW is adopted as base optimizer. The hyperparameters can be found in Table 11. Our results are obtained from three random trials. Each trial takes roughly 8 hours on our hardware.

Table 11: Hyperparameters used for GPT2.

| Hyper-parameters | Values |
|---|---|
| LoRA $r$ (rank) | 4 |
| LoRA $\alpha$ | 32 |
| # epochs | 5 |
| batchsize | 8 |
| learning rate | $2 \times 10^{-4}$ |
| label Smooth | 0.1 |
| $\mu_0$ for BAR | 0.1, 0.15, 0.2, 0.25, 0.3 |
| scheduler for BAR | linear, constant |
| beam size | 10 |
| length penalty | 0.8 |

