# OpenReview forum: "Implicit Regularization of Sharpness-Aware Minimization for Scale-Invariant Problems"
_NeurIPS.cc/2024/Conference — NeurIPS 2024 poster_

### Official Review · Reviewer_t2ze · 2024-07-01

**Soundness:** 4
**Presentation:** 4
**Contribution:** 4
**Rating:** 8
**Confidence:** 3

**Summary:**

This paper analyzes the implicit regularization effect of Sharpness-Aware Minimization (SAM), focusing specifically on scale-invariant problems. While existing research emphasizes sharpness, this study introduces a new concept called Balancedness, demonstrating both theoretically and empirically that SAM promotes Balancedness. Additionally, the proposed Balancedness-Aware Regularization (BAR) is shown to significantly improve computational efficiency while achieving superior performance in fine-tuning language models using LoRA.

**Strengths:**

Originality
- Introducing a new concept of Balancedness and reinterpreting the implicit regularization effect of SAM from a new perspective.

Quality
- Consistent theoretical analysis and empirical validation with clear results.

Clarity
- Clear presentation with a structure that is easy for readers to understand.

Significance
- Revealing a new aspect of SAM's adaptability to data anomalies and proposing a computationally efficient BAR.

Soundness
- The technical claims, experimental methods, and research techniques of this paper are robust, and the central claims are well-supported by ample evidence. Both the theoretical analysis and experimental results are consistent, and the conclusions are clear. Potential concerns, such as those regarding m-sharpness, are addressed.

Contribution
- The paper theoretically analyzes the mechanism by which SAM promotes balance and proves that balance converges to |Bt| \rightarrow 0. It particularly highlights the strong impact of data anomalies (outliers). This explains why SAM outperforms SGD even in the presence of data anomalies. The implicit regularization of SAM is made explicit, and a new, computationally efficient variant of SAM called Balancedness-Aware Regularization (BAR) is proposed.
- Experiments using various models confirm that the proposed BAR exhibits superior performance in fine-tuning with LoRA compared to traditional SAM and SGD.
- While SAM has been noted as a promising optimization method to enhance the performance of large language models [1], its high computational cost and implementation complexity have limited its practical use. However, this study's regularization from the perspective of Balancedness has a potentiall to addresses these issues in conjunction with LoRA.

[1] https://arxiv.org/abs/2210.14199

**Weaknesses:**

- While LoRA training is performed with AdamW in the original paper, demonstrating that SAM or BAR is more effective than these would make the paper more solid.
- Code such as `eval.py` was not included in the supplementary materials, and its release is expected.

**Questions:**

- As cited in the Appendix, it is known that SAM promotes the acquisition of low-rank features [2]. However, it was pointed out that explicit regularization for acquiring low-rank features did not lead to improved generalization ability. My curiousity is about the relationship between Balancedness and low-rank features, and whether there is any connection to sharpness.

- Another question is how optimization methods commonly used in LLM training, such as AdamW, affect Balancedness.

[2] https://arxiv.org/abs/2305.16292

**Limitations:**

The authors sufficiently discuss the limitations of this study, particularly regarding the application range and computational efficiency of BAR.

---

> ### Author Rebuttal · Authors · 2024-08-03
>
> We appreciate the time and efforts from the reviewer put into this review. We also want to thank the reviewer for recognizing the strength of our work. We will update our draft and code repo to further improve the quality of this work.
>
> **W1.** *While LoRA training is performed with AdamW in the original paper, demonstrating that SAM or BAR is more effective than these would make the paper more solid.*
>
> The LoRA baselines in our experiments are indeed trained with AdamW (or Adam for different experiments), and our implementation follows the official LoRA repo.
>
> **W2.** *Code such as ```eval.py``` was not included in the supplementary materials, and its release is expected.*
>
> The code for ```eval.py``` can be found in the official repo of LoRA; see [1]. We will update the ReadMe.md to provide improved instructions.
>
> **Q1.** *As cited in the Appendix, it is known that SAM promotes the acquisition of low-rank features [2]. However, it was pointed out that explicit regularization for acquiring low-rank features did not lead to improved generalization ability. My curiousity is about the relationship between balancedness and low-rank features, and whether there is any connection to sharpness.*
>
> The setting in [2] does not extend to our case. This is because that the feature learning happens in the principle space in [2], while LoRA learns features in the residue space. Recall that low rankness is induced by parameterizing the last layer as $\mathbf{A} \mathbf{B}^\top$; see section 6 of [2]. In other words, the learned feature is $\mathbf{X}\mathbf{A} \mathbf{B}^\top$, where $\mathbf{X}$ is data or feature from the previous layer.  However, in our LoRA case, the learned feature is $\mathbf{X}(\bf{W} + \mathbf{A} \mathbf{B}^\top ) $, where $\mathbf{W}$ is the pretrained model. In sum, low rankness can play fundamentally different roles in [2] compared with our LoRA case.
>
> Regarding the relation of balancedness and sharpness, there are indeed links such as what we have shown in Lemma 1. Regarding the relation between balancedness and low rank features, this is beyond the scope of this work. As there are multiple approaches to induce low-rankness (and [2] employs one of them), we believe that a more systematic study should be carried out for answering this question, and we leave it to future work.
>
>
> **Q2.** *Another question is how optimization methods commonly used in LLM training, such as AdamW, affect Balancedness.*
>
> It is not difficult to analytically show that a variant of ADAM (without momentum) decreases balancedness slightly on a simple loss function $f(x,y)=0.5(xy - 1)^2$ when learning rate $\eta \rightarrow 0$. However, the balancedness does not decrease after approaching near optimal. This can be seen from Fig. 1 in the additional PDF file (see general response). Note that we use $\eta = 5e-4$ with $10^4$ iterations. However, this analysis on the simple loss function may not extend to more general cases.
>
> In practice, we perform the same experiments as Fig. 3 and plot the balancedness of all layers after training using Adam in Fig. 2 of the additional PDF. No specific trend on balancedness (2${\cal B}_{t,l}$) is observed, because the balancedness on some layers increases compared to initialization; while decreases for other layers. Note that the initialized balancedness is within (2.7 - 3.2) for all layers.
>
> **References**
>
> [1] LoRA official repo: https://github.com/microsoft/LoRA/blob/main/examples/NLG/eval/eval.py
>
> [2] M. Andriushchenko, D. Bahri, H. Mobahi, N. Flammarion. Sharpness-aware minimization leads to low-rank features. NeurIPS 2023

---

> > ### Comment · Reviewer_t2ze · 2024-08-08
> > **Official Comment by Reviewer t2ze**
> >
> > I thank the authors for their detailed response. I am satisfied with the answers provided, and I would like to keep my positive score.

---

> > > ### Author Response · Authors · 2024-08-12
> > > **Thank you for your comments**
> > >
> > > Thank you once again for acknowledging the strength of our work. We will revise our manuscript to incorporate your suggestions and include additional instructions in the ReadMe.md file.

---

### Official Review · Reviewer_9YQw · 2024-07-12

**Soundness:** 4
**Presentation:** 3
**Contribution:** 4
**Rating:** 7
**Confidence:** 3

**Summary:**

This paper investigates the dynamics of SAM when the loss is of the form f(xy^T) or f(x^Ty). This formulation includes interesting scenarios such as LoRA. This paper shows that SAM will promote balancedness, which is the difference between the squared norms of two variables. Based on this new analysis, this paper proposes to regularize balancedness and show that this method leads to similar or sometime better result than SAM with significantly less compute.

**Strengths:**

1. This paper considers the interesting interplay between SAM and scale-invariance of the loss and discovers that SAM implicitly regularizes balancedness in this case.
2. The empirical verification of the theoretical prediction is thorough and convincing.

**Weaknesses:**

1. Arguments including that SGD will make balancedness unchanged are only correct for infinitesimal learning rate and claims like this should be made rigorous (for example at line 137).

2. In section 2.1, when the limitation of sharpness is discussed, the example that h(x,y) = xy is very confusing because there is no local minima for such loss.

**Questions:**

1. Because SGD with a small learning rate will make balancedness almost unchanged, does the result imply that initializing LoRA to be completely balanced and using SGD should also improve performance?

2. Connecting to the first problem, as in Figure 3, the balancedness of the weight trained by SAM, while decreasing faster than SGD, still remains at a high level. Is this the case for BAR as well?

**Limitations:**

The limitation is clearly addressed.

---

> ### Author Rebuttal · Authors · 2024-08-03
>
> We appreciate the reviewer for the nice questions. We will add these discussions to the draft.
>
> **W1.** *Arguments including that SGD will make balancedness unchanged are only correct for infinitesimal learning rate and claims like this should be made rigorous (for example at line 137).*
>
> Thank you for pointing this out. We will proofread a few more times and rephrase these sentences. In Figure 3, we have shown that even with a learning rate $0.1$, SGD only slightly changes the balancedness.
>
> **W2.** *In section 2.1, when the limitation of sharpness is discussed, the example that $h(x,y) = xy$ is very confusing because there is no local minima for such loss.*
>
> Thank you for pointing this out. We will rephrase this sentence. Our intention was that the function $h(x,y) = xy$ has the same Hessian (hence eigenvalues) for any $(x, y)$. Thereby, one cannot provide any implicit bias using the largest eigenvalue.
>
>
> **Q1.** *Because SGD with a small learning rate will make balancedness almost unchanged, does the result imply that initializing LoRA to be completely balanced and using SGD should also improve performance?*
>
> Unfortunately, it is difficult to initialize LoRA in a balanced fashion. Using the notation of equation (6), one has to set $\mathbf{Y} = 0$ to ensure that at initialization $\mathbf{X}\mathbf{Y}^\top = \mathbf{0}$. To achieve exact balancedness, the only choice is to choose $\mathbf{X}=0$ as well. This is not a good initialization because $\mathbf{X}=0$ and $\mathbf{Y}=0$ is a saddle point (as one can see from the gradients).
>
> An alternative is to initialize $\mathbf{Y}=0$ and choose a small $\mathbf{X}$ in magnitude, i.e., initializing close to a saddle point. However, it is known that SGD escapes saddle points quite slow [1].
>
> This means that LoRA initialized in a balanced manner can slow down convergence, and it in turn necessities our balancedness-aware regularization (BAR).
>
> **Q2.** *Connecting to the first problem, as in Figure 3, the balancedness of the weight trained by SAM, while decreasing faster than SGD, still remains at a high level. Is this the case for BAR as well?*
>
> In Figure 3, the balancedness of SAM is decreasing less faster because of the multiple layers of the RoBERTa. As we have discussed in lines 202 - 206 and Theorem 5 in Appendix, multiple layers slowdown the implicit regularization of SAM, and the slowdown is roughly proportional to the square root of (LoRA) layers. In the case of Figure 3, there are 24 transformer layers, and LoRA is applied to every query and value in each transformer layer. This gives a $\sqrt{24 * 2} \approx 7$x slowdown.
>
> The proposed BAR does not suffer from the slowdown from number of layers, because the regularization can be applied individually on each LoRA module for the query and value layers. This gives a better regularization on balancedness and explains why nBAR can improve over SAM in Table 3.
>
>
> **References**
>
> [1] C Fang, Z Lin, and T Zhang. Sharp analysis for nonconvex SGD escaping from saddle points. COLT 2019.

---

> ### Comment · Reviewer_9YQw · 2024-08-07
>
> Thank the authors for the detailed response. Regarding the initialization of LoRA, it does not appear natural to me that the initialization must be function-preserving (see for example [1]) but this seems beyond the scope of this paper. I think Theorem 5 is very interesting as an explanation for the small decrease in balancedness. I will increase my score to 7.
>
> [1] LoRA-GA: Low-Rank Adaptation with Gradient Approximation https://arxiv.org/pdf/2407.05000

---

> ### Author Response · Authors · 2024-08-12
>
> Thank you for your careful consideration and for recognizing the strengths of our work.
>
> The balancedness observed in the LoRA-GA paper might indeed contribute to their strong empirical performance. While we also want to point out that balanced initialization can sometimes lead to saddle points, and therefore how to initialize seems to be quite problem-dependent.  Consider a simple two-dimensional problem $(xy - 1)^2$ for example. A balanced initialization $(c, -c)$ for any constant $c$ drives GD to stuck on saddle points (0, 0). This can be seen by writing down the gradients.
>
> We sincerely thank the reviewer once again for the nice suggestions. We will update our manuscripts accordingly to further improve its quality.

---

### Official Review · Reviewer_7PUq · 2024-07-13

**Soundness:** 3
**Presentation:** 3
**Contribution:** 3
**Rating:** 7
**Confidence:** 3

**Summary:**

This paper investigates the implicit regularization effects of sharpness-aware minimization (SAM) on scale-invariant optimization problems, introducing "balancedness" as a new metric for analysis. The authors provide theoretical results showing that SAM promotes balanced solutions for both non-overparameterized and overparameterized scale-invariant problems, and demonstrate that SAM's regularization effect is stronger on noisy data. Based on these insights, they propose balancedness-aware regularization (BAR), a computationally efficient variant of SAM. The paper evaluates BAR on language model fine-tuning tasks using LoRA, demonstrating that it can match or exceed SAM's performance while significantly reducing computational overhead.

**Strengths:**

- The theoretical analysis provides new insights into SAM's implicit regularization effects by introducing "balancing".
- The proposed BAR method offers a more efficient alternative to SAM for scale-invariant problems by adding explicit regularization.
- Comprehensive experiments on language model fine-tuning demonstrate practical benefits.

**Weaknesses:**

- The scope is limited to scale-invariant problems, and LoRA types optimization problem.
- Assumption 1 requires  Lipschitz continuous gradient which may not always hold in practice (e.g., ReLU network)

**Questions:**

- How to understand the lower bound in Theorem 3, which is not always greater than zero during the optimization.

- How to choose a good parameter $\rho$?

**Limitations:**

Yes, the authors adequately addressed the limitations.

---

> ### Author Rebuttal · Authors · 2024-08-03
>
> We thank the reviewer for the time and efforts devoted to this paper. Below please find our responses to weakness and questions.
>
> **W1.** *The scope is limited to scale-invariant problems, and LoRA types optimization problem.*
>
> We agree with the reviewer that the major application is on scale-invariant problems such as LoRA and its variants. However, we hope to emphasize that LoRA is already one of the most popular approaches for finetuning language models. LoRA is more economical and easier to serve in practice compared to full parameter-tuning. LoRA type approaches are actively developed and well welcomed by the community; see e.g., HuggingFace's PEFT codebase [1]. Given these evidences, we believe that this scenario is important and our results are useful.
>
> Moreover, we definitely hope to inspire following-ups and even entirely new approaches to extend our results to more general settings. We are positive that this will take place in a step-by-step fashion, and we hope that our work is helpful for initiating this.
>
> **W2.** *Assumption 1 requires Lipschitz continuous gradient which may not always hold in practice (e.g., ReLU network).*
>
> Lipschitz continuous gradient (or smoothness) is a standard assumption for the theoretical understanding of gradient-based methods; see e.g., SGD [2], SGD with momentum [3], AdaGrad [4], and Adam [5]. While smoothness does not necessarily hold in every practical scenario, the theoretical insights from the analysis still yield robust and meaningful guidance for practice, evidenced by the popularity of [2 - 5] for training neural networks.
>
> Moreover, we also hope to point out that in our LoRA scenario, there is even no ReLU activation for the trainable parameter (activations only apply for the frozen parameters). Thus, smoothness is slightly more reasonable compared to the case of Adam on deep neural networks.
>
>
> **Q1.** *How to understand the lower bound in Theorem 3, which is not always greater than zero during the optimization.*
>
> When it is smaller than $0$, the inequality trivially holds. However, when $\rho$ is chosen small, it is more likely that the RHS is positive given that $\rho^2 \ll \rho$. We will update the lower bound to $\max$ {0, lower bound in Theorem 3}.
>
> **Q2.** *How to choose a good parameter $\rho$?*
>
> Empirical experiences recommend $0.05$ or $0.1$. In practice, a grid search is also helpful.
>
>
> **References.**
>
> [1] https://github.com/huggingface/peft
>
> [2] S Ghadimi, and G Lan. Stochastic First- and Zeroth-order Methods for Nonconvex Stochastic Programming. SIAM J-OPT 2013.
>
> [3] Y Liu, Y Gao, and W Yin. An Improved Analysis of Stochastic Gradient Descent with Momentum. NeurIPS 2020
>
> [4] R Ward, X Wu, and L Bottou. AdaGrad Stepsizes: Sharp Convergence over Nonconvex Landscapes. JMLR 2020
>
> [5] X Chen, S Liu, R Sun, and M Hong. On the Convergence of A Class of Adam-Type Algorithms for Non-Convex Optimization. ICLR 2019

---

> > ### Comment · Reviewer_7PUq · 2024-08-12
> >
> > Thank you for the detailed rebuttal.  As my main concerns are addressed, I will increase my score to 7.

---

> > > ### Author Response · Authors · 2024-08-13
> > >
> > > Thank you for your thoughtful consideration. We appreciate your feedback and are glad that we could address your concerns.

---

### Official Review · Reviewer_Cq3N · 2024-07-23

**Soundness:** 3
**Presentation:** 2
**Contribution:** 4
**Rating:** 6
**Confidence:** 5

**Summary:**

### Summary:

This work introduces balancedness (instead of sharpness) and proposes Balancedness-Aware Regularization (BAR), used for scale-invariant problems (e.g., LoRA). Given an objective such as $f(xy^T)$ or $f(x^Ty)$ with parameters $x,y$ and a fixed function $f$, the balancedness is defined as $B_t = \frac{1}{2}(\|x_t\|^2 - \|y_t\|^2)$. They prove that SAM decreases balancedness while SGD preserves it (Theorems 1&2&3) and argue that this new quantity goes beyond local information and relies on the entire SAM trajectory. Indeed, they show that whether $B_t$ goes to zero and gets quite small under SAM.
Moreover, they show that outliers strongly affect balancedness, which helps explain SAM behavior. Finally, they propose the BAR algorithm and conclude with several experiments.

**Strengths:**

### Pros:

- the studied problem is very important, and the result can be impactful
- the balancedness is completely new and well-motivated for SAM (previously some works studied related quantities but just for SGD)
- having several experiments

**Weaknesses:**

### Cons:

- the paper is not well-written, and it is difficult to follow it. The authors need to explain results, ideas, and notation via simple sentences and then deal with formulae/equations

- some ambiguities about Theorem 1

**Questions:**

### Questions/Comments:


This is an interesting paper. The authors nicely mention that balancedness changes under SAM and this could be one potential way to explain SAM behavior. Moreover, they propose balancedness minimization, which leads to even better algorithms. Unfortunately, the paper must be modified to reach the audience and become readable since the current version is hard to follow.

Some comments:

- line 37-39 -- this part is not well written and is confusing. What is the role of $d_1+d_2$? I suggest explaining it in a few sentences in the paper.

- Limitation of sharpness (Section 2.1): the max e.v. is not the only sharpness measure considered in the literature. Indeed, there is apparently a class of invariant sharpness measures for SAM; for example, see this:
        - A Universal Class of Sharpness-Aware Minimization Algorithms, ICML 2024

- Do authors claim that Theorem 1 is the original result of this paper? I think this is known from previous works on the implicit biases of SGD, but it looks like the paper is claiming the results without references. See, for example, Equation 2.5 in this paper (already cited):
        - Ji, Ziwei, and Matus Telgarsky. Gradient descent aligns the layers of deep linear networks. ICLR.

Please explain whether the theorem follows from the previous results or if there is a major difference between them.

 - The BAR algorithm is barely discussed in the main body of the paper—the algorithm itself is provided in the appendix. Because it is one of the main contributions of the paper, authors should consider rearranging stuff in the paper to place it in the main body and discuss it more.

- Moreover, the ideas behind Algorithms 2 and 3 should be discussed in the main body because of their importance. What are the role of the hyperparameters in the algorithms?

---

> ### Author Rebuttal · Authors · 2024-08-03
>
> We thank the reviewer for the time and efforts devoted to this paper. Below please find our point-to-point responses. Please let us know if there are other questions or unclearness during discussion. We are always glad to improve the quality of our submission.
>
> **W1.** *The paper is not well-written, and it is difficult to follow it.*
>
> We hope a brief summary of this paper helps to clarify the overall logistic.
>
> * We show that SAM promotes balancedness on scale-invariant problems. Unlike many of existing metrics, balancedness is a global metric for implicit regularization. Theoretical and empirical evidences are provided to support SAM's implicit regularization on balancedness.
>
> * Since balancedness is computationally tractable, we mimic the dynamic of SAM (e.g., Theorem 2) and explicify balancedness as a regularizer -- BAR. BAR can be applied on top of other optimizers such as SGD or Adam in the same way as weight decay. BAR overcomes the need for the second gradient computation in SAM, yet achieving similar numerical performance as SAM.
>
> We are also glad to clarify any ambiguities and revise the draft for better presentation -- please let us know during the discussion phase.
>
>
> **W2.** *Some ambiguities about Theorem 1*
>
> Since the confusion is detailed in questions, see our responses in Q3.
>
>
> **Q1.** *line 37-39... What is the role of $d_1 + d_2$?*
>
> Taking NOP problem $\min_{\mathbf{x}, \mathbf{y}}f(\mathbf{x}\mathbf{y}^\top)$ with $\mathbf{x} \in R^{d1}$ and $\mathbf{y} \in R^{d2}$ as an example. Here, we have that the loss function $f : R^{d1 \times d2} \mapsto 1$, in other words, the dimension of $dom f$ is $d1 \times d2$. The number of variables to be optimized is only $dim(\mathbf{x}) + dim(\mathbf{y}) = d1 + d2$ since we parametrize $dom f$ by $\mathbf{x}\mathbf{y}^\top$. Since the number of variables is smaller than the dimension of $dom f$, we are using insufficient variables to perform optimization, and hence its name -- non-overparametrization.
>
>
> **Q2.** *Other sharpness ... There is apparently a class of invariant sharpness measures for SAM; for example, see this: - A Universal Class of Sharpness-Aware Minimization Algorithms, ICML 2024*
>
> Thank you for pointing out this work. Unfortunately, it is impossible for us to cite this ICML paper because it appeared later on arXiv (June 6) than NeurIPS submission due date (May 22). However, we have cited and discussed the differences with an earlier workshop version of the same ICML paper; see lines 596 - 597 in Appendix. Per request, the main differences with the ICML paper are re-summarized below.
>
> The ICML paper introduces generalized sharpness measures -- sharpness as any function of eigenvalues of Hessian. However, even the generalized sharpness cannot provide implicit regularization for function $h(x,y)=xy$, simply because the Hessian is the same for all $(x, y)$. In addition, when Hessian is negative definite, some of the generalized sharpness measures (e.g., determinate of Hessian) may not be necessarily meaningful. Balancedness overcomes these two problems.
>
> In terms of practical algorithms inspired by theoretical derivations, our goal is in sharp difference with the ICML paper. Our balancedness targets at computational efficiency of SAM. On the other hand, the ICML paper targets at improving over SAM under the same or slightly more computational budget (when the number of samples $n$ is set to be greater than $1$ in their Algorithm 1).
>
>
> **Q3.** *Do authors claim that Theorem 1 is the original result of this paper? I think this is known from previous works on the implicit biases of SGD, but it looks like the paper is claiming the results without references. See, for example, Equation 2.5 in this paper (already cited): - Ji, Ziwei, and Matus Telgarsky. Gradient descent aligns the layers of deep linear networks. ICLR.*
>
> There is a potential misunderstanding. We do not hope to take credits from Theorem 1, as we have already cited several previous works in lines 131 - 134. Even in the ICLR paper of [Ji and Matus 2019], they have cited the same set of papers as we do. We slightly extend the results from these cited papers to the case with stochastic gradients. We will rephrase this paragraph to eliminate possible misunderstanding.
>
>
> **Q4 \& Q5.** *The BAR algorithm is barely discussed in the main body of the paper—the algorithm itself is provided in the appendix ... Moreover, the ideas behind Algorithms 2 and 3 should be discussed in the main body because of their importance. What are the role of the hyperparameters in the algorithms?*
>
> The design and the implementation of balancedness-aware regularizer (BAR) are already discussed in lines 280 - 285 in the main context. Since BAR is a regularizer, it can be used the same way as e.g., an $\ell_2$ regularizer or weight decay. Note that this is quite standard, hence we assume the use of BAR should also be straightforward. Algs. 2 and 3 are just detailed BAR implementation in the same format as weight decay. Given the space limitation, instead of duplicating this weight-decay type implementation in the main context, we provide a short yet informative sentence in line 284 - 285 that convey the same massage. However, we would love to expand our discussions if we had additional space.
>
> The ideas of Algs. 2 and 3 have been already discussed in lines 274 - 278 and lines 280 - 282, respectively. To summarize, the idea is that since SAM promotes balancedness implicitly (e.g., in Theorem 2),  we can explicify this dynamic and use it to regularize balancedness.
>
> The only hyperparameter for BAR is the coefficient on this regularizer. It plays the same role as the coefficient for e.g., an $\ell_2$ regularizer, that is, determining the tradeoff between regularization and loss-fitting.

---

> > ### Comment · Reviewer_Cq3N · 2024-08-12
> >
> > Dear authors, thank you for your detailed response. I have a few follow-up questions:
> >
> > - (Q3) what is your plan on Theorem 1? Could you detail how you want to change its presentation?
> >
> > - (Q4&5) you will have more space for the final version, What do you plan to do about the explanation of the algorithm?

---

> ### Author Response · Authors · 2024-08-12
>
> We appreciate the reviewer's suggestions regarding the presentation and clarity. We hope that the revisions we have made in response to your comments will improve the clarity of our work. Here's our detailed plans for the revision.
>
> **Q3.** Our objective is to rephrase lines 131-134 to clearly indicate that these results have been documented in previous studies, including those by e.g., Aurora et al. (2018, 2019b), Ji and Telgarsky (2019), and Ahn et al. (2023). Additionally, we will explicitly reference these papers in Theorem 1.
>
> An example of the revision could be:
>
> > How does ${\cal B}_t$ evolve in different algorithms? To set a comparing benchmark of SAM, we first **borrow** results in (Aurora et al 2018, 2019b; Ji and Telgarsky, 2019; Ahn et al 2023).
> >
> > Theorem 1. (**[Aurora et al 2018, 2019b; Ji and Telgarsky, 2019; Ahn et al 2023].**) When applying SGD on the NOP (1a), the ...
>
>
> **Q4 \& 5.** As suggested, our plan is to provide additional details and intuition to enhance understanding. We will ensure that our revision is presented in a clear and accessible manner. Our plans include, but are not limited to, the following:
>
> - To move Algs. 2 and 3 into the main body. We hope that this can make the sentences like 'BAR can be implemented in the same manner as weight decay' more concrete.
>
> - To include a dedicated paragraph, with a bold subtitle, to explain the ideas and intuitions behind BAR. We will expand lines 274 - 278 and 280 - 282 to illustrate how Theorems 2 and 3 can be adapted to derive BAR.
>
> - To explain Fig. 2(b) in more depth to show that our BAR indeed mimics SAM's dynamics.
>
> - To discuss the only hyperparameter of BAR and its role in balancing the trade-off between regularization and loss fitting.

---

> > ### Comment · Reviewer_Cq3N · 2024-08-12
> >
> > I appreciate the authors' detailed plan for the next revision of the paper. My comments and concerns have now been mostly addressed. For the accepted version of the paper, I ask the authors to ensure that they apply the promised changes (Q3, Q4, Q5 above). I decided to increase my score, provided the authors make the promised changes. Good luck!

---

> > > ### Author Response · Authors · 2024-08-13
> > >
> > > We will ensure that the revisions enhance presentation following what we have discussed above. Thank you for the suggestions to improve the quality of this work!

---

### Author Rebuttal · Authors · 2024-08-03

We thank the ACs and reviewers for handling this submission. Your comments are appreciated, and the manuscript will also be updated accordingly. Our point-to-point responses can be found below, and a .pdf file is also attached as graphical illustration to questions from Reviewer t2ze.

---

### Decision · Program_Chairs · 2024-09-25

**Decision:**

Accept (poster)

**Comment:**

The paper presents a novel and well-supported contribution to the understanding and application of SAM in scale-invariant problems, particularly in the context of LoRA. The introduction of BAR (Balancedness-Aware Regularization) offers a practical and computationally efficient alternative to SAM, with potential for significant impact in the field. The reviewers unanimously recommended to accept this paper.

For the camera-ready version, the authors should incorporate suggestions and discussions in the review and rebuttal, including improving the clarity and presentation of the BAR algorithm and its theoretical foundations as well as explicitly addressing and clarifying any assumptions made in the paper.